# A biotin targeting chimera (BioTAC) system to map small molecule interactomes in situ

Andrew J. Tao [1,5], Jiewei Jiang [1,5], Gillian E. Gadbois[1], Pavitra Goyal [1], Bridget T. Boyle[1], Elizabeth J. Mumby[2], Samuel A. Myers [3], Justin G. English [2] ✉ & Fleur M. Ferguson [1,4] ✉

Understanding how small molecules bind to specific protein complexes in living cells is critical to understanding their mechanism-of-action. Unbiased chemical biology strategies for direct readout of protein interactome remodelling by small molecules would provide advantages over target-focused approaches, including the ability to detect previously unknown ligand targets and complexes. However, there are few current methods for unbiased profiling of small molecule interactomes. To address this, we envisioned a technology that would combine the sensitivity and live-cell compatibility of proximity labelling coupled to mass spectrometry, with the specificity and unbiased nature of chemoproteomics. In this manuscript, we describe the BioTAC system, a small-molecule guided proximity labelling platform that can rapidly identify both direct and complexed small molecule binding proteins. We benchmark the system against μMap, photoaffinity labelling, affinity purification coupled to mass spectrometry and proximity labelling coupled to mass spectrometry datasets. We also apply the BioTAC system to provide interactome maps of Trametinib and analogues. The BioTAC system overcomes a limitation of current approaches and supports identification of both inhibitor bound and molecular glue bound complexes.

Small molecule targeted therapies are a cornerstone of modern medicine, providing effective treatments for a wide range of diseases. However, the clinical effects of candidate therapies in patients can be highly variable, even when they bind the same therapeutic target with equal affinity[1]. This divergent pharmacology may be due to uncharacterized off-target effects, combinatorial polypharmacology, or from small molecules inducing changes in the interactome of their protein targets through allosteric effects or molecular glue effects[2]. Despite the critical role of small molecules in drug discovery and development, there is a lack of comprehensive, network-scale profiling methods that inform on the cellular interactomes of small molecules.

Examples of small-molecule mediated interactome rewiring are well studied in cancer, where targeted therapies frequently induce functional changes in the complexation of their protein targets[3]. For example, blockbuster cancer drugs, such as trametinib[1] and lenalidomide[4–6], exert efficacy through the promotion of novel protein complexes, now known as 'molecular glue' pharmacology. Unanticipated protein complex rewiring is also a major cause of drug candidate failures, for example underpinning adverse effects of 1st generation RAF inhibitors in RAS driven tumors[7], and resistance to BET bromodomain inhibitors in triple-negative breast cancer and neuroblastoma[8,9]. However, the discovery of interactome changes that mediate both drug efficacy and drug resistance has so far been

[1]Department of Chemistry and Biochemistry, University of California San Diego, La Jolla, CA 92093, USA. [2]Department of Biochemistry, University of Utah School of Medicine, Salt Lake City, UT 84112, USA. [3]Laboratory for Immunochemical Circuits, La Jolla Institute for Immunology, La Jolla, CA 92037, USA. [4]Skaggs School of Pharmacy and Pharmaceutical Sciences, University of California San Diego, La Jolla, CA 92093, USA. [5]These authors contributed equally: Andrew J. Tao, Jiewei Jiang. ✉e-mail: justin.english@biochem.utah.edu; fmferguson@ucsd.edu

serendipitous, as researchers investigate why certain drugs display unexpected pharmacology in the clinic. Despite their central importance, effects on target complexation remain uncharacterized for most protein ligands, representing a 'blind spot' in compound characterization workflows.

Current gold-standard technologies for unbiased target-ID are the chemical proteomic techniques photoaffinity labeling[10] and micro-environment mapping (μMap)[11] that use UV-light or blue light initiated diazirine photochemistry to label liganded proteins with affinity handles. In chemical proteomic experiments, off-compete conditions with excess of free drug, or comparison to a stereoisomeric negative control molecule facilitate identification of true hits from non-specific noise[11]. However, in the intracellular context these techniques are directed towards detection of the primary target(s) only, due to the short half-life of the generated reactive carbene species which corresponds to a labeling radius of ~4 nm[11]. Here, the linker length between the diazirine or iridium photocatalyst and the drug dictates the labeling radius, and is therefore limited by cell permeability of the conjugate[11]. As such, they have yet to be applied to map drug-bound complexes. Diazirine photochemistry approaches are also currently incompatible with in vivo applications.

To understand drug-induced interactome changes, affinity purification coupled to mass spectrometry (AP/MS) and proximity labeling coupled to mass spectrometry (PL/MS) are commonly employed[12]. Proximity labeling techniques such as BioID and TurboID are particularly advantageous in mapping interactomes[13]. Proximity labeling methods have a labeling radius of up to 35 nm and can be used in live cells and in vivo[14]. By fusing a target protein or localization tag with a proximity labeling enzyme, proximity labeling can reliably detect transient, moderate affinity protein-complex interaction in situ due to the ability of the biotin ligase to accumulate affinity tags on these

protein partners over time[15]. However, both techniques rely on a priori target knowledge, which is often incompletely understood for drug candidates, and the fusion of the target protein to either an affinity-tag or a proximity labeling enzyme, which can significantly impact interacting proteins[15]. Therefore, they cannot be used for unbiased interactome-ID of small molecules. The outputs of these techniques are also typically large numbers of enriched proteins, due to their low stringency, making data interpretation and validation challenging.

To facilitate routine evaluation of ligand-target interactome changes induced by either inhibitors or molecular glues in a single experiment, we envisioned a method that combines the precision and unbiased nature of chemical proteomics with the sensitivity, whole-organism compatibility, and extended detection radius of proximity-labeling coupled to mass spectrometry. Here, we report the development of a ligand-guided miniTurbo method to accomplish these goals, and benchmark it against gold-standard unbiased technologies for both target-ID and interactome-ID.

## Results

Our method, named the biotin targeting chimera (BioTAC) system, uses bifunctional molecules composed of a compound-of-interest linked to selective FKBP12F36V recruiter orthoAP1867, to recruit a ligandable proximity labeling enzyme (miniTurbo-FKBP12F36V) to compound-bound complexes enabling their biotinylation and subsequent affinity purification (Fig. 1A, B). To benchmark the BioTAC system for accurate detection of the direct targets of ligands we selected the well-characterized BET protein inhibitor (+)-JQ1, which potently binds to the bromodomains of BRD2, of BRD3 and of BRD4 (as well as the testes-specific BET protein BRDT) as a test case[15,16]. We synthesized a series of bifunctional molecules consisting of (+)-JQ1 conjugated to orthoAP1687 via a variable linker (Fig. 1B,

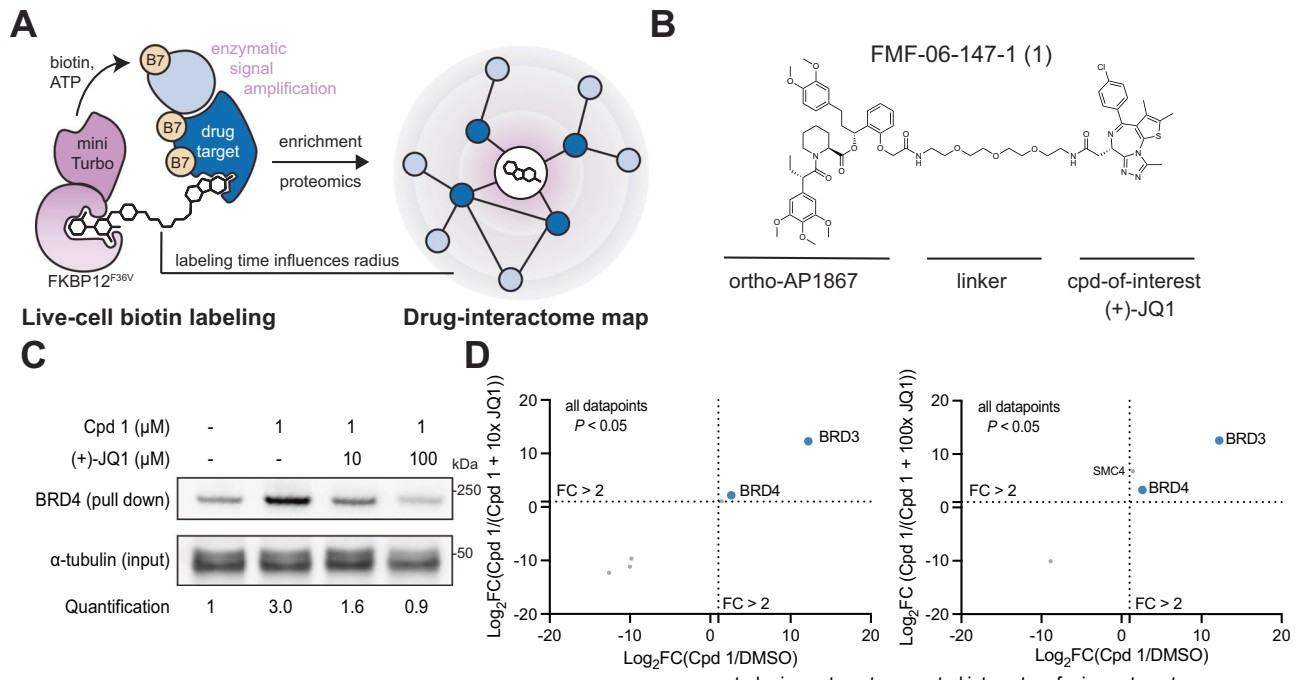

**Fig. 1 | The BioTAC system enables rapid, accurate small molecule target-ID.** **A** Schematic depicting the components of the BioTAC system. **B** Example molecule, Cpd **1**, annotated with key functional groups. **C** Immunoblot analysis of BRD4 enrichment following treatment of HEK293 cells transiently transfected with miniTurboFKBP12F36V with the indicated compounds and 100 μM biotin at the 30 min timepoint (input = sample processing control, n = 3 biological replicates, Supplementary Fig. 1I, J). **D** Scatterplot displaying relative FC of streptavidin-enriched protein abundance following treatment of HEK293 cells transiently transfected with miniTurboFKBP12F36V with either DMSO or 1 μM Cpd **1** plus the indicated concentration of (+)-JQ1 competitor and 100 μM biotin at the 30 min timepoint. Only proteins with P-value < 0.05 (two-sample moderated T test) in both conditions depicted. FC fold-change, complete datasets in Supplementary Data 1, plotted individually in Supplementary Fig. 4. Source data are provided as a Source Data file.

Supplementary Fig. 1A–C). We measured (+)-JQ1-bifunctional molecule cell permeability using an FKBP12[F36V] cellular target engagement assay adapted from Nabet et al.[17]. Briefly, FKBP12[F36V] binding molecules compete with an FKBP12[F36V] targeting PROTAC (dTAG-13), for NLuc-FKBP12[F36V] occupancy, rescuing degradation and resulting in an increase in NLuc/FLuc (control) signal relative to DMSO-treated cells[17]. All synthesized compounds were comparably cell permeable (Supplementary Fig. 1D–F)[17].

We next performed bifunctional molecule guided-proximity labeling experiments using our validated reagents. HEK293 cells transiently transfected with miniTurbo-FKBP12[F36V] were treated with 100 μM biotin, 1 μM bifunctional (+)-JQ1 recruiters, and variable concentrations of free (+)-JQ1 to off-compete the bifunctional molecule and rescue biotinylation. Biotinylated proteins were isolated from cell lysates via streptavidin bead pulldown and analyzed by western blot for BRD4, a primary target of (+)-JQ1. We observed significant enrichment of BRD4 and a dose dependent decrease in BRD4 pulldown in the presence of unconjugated (+)-JQ1 at all timepoints evaluated (Fig. 1C, Supplementary Fig. 1G–J). Comparable activity was observed for all bifunctional analogues, indicating low sensitivity to linker length (Supplementary Fig. 1G–J), FMF-06-147-1 (Cpd 1) was selected for further characterization based on high BRD4-labeling and rescue at the 30 min time point (Fig. 1C).

To confirm that labeling was occurring in the biologically relevant compartment (the nucleus), we imaged cells transfected with HA-tagged miniTurbo-FKBP12[F36V], and treated with DMSO, 1 μM Cpd 1, or 1 μM Cpd 1 + 10 μM (+)-JQ1. In the absence of Cpd 1, miniTurbo-FKBP12[F36V] is predominantly localized in the cytosol. Upon addition of Cpd 1, miniTurbo-FKBP12[F36V] translocates to the nucleus, and co-localizes with BRD4, in a manner that depends on BRD4 binding (Supplementary Fig. 2). This small-molecule induced nuclear localization of FKBP12[F36V] by BRD4-targeting bifunctional molecules is consistent with a recent paper by Gibson et al., that reported identical effects with FKBP12[F36V]-tagged GFP constructs[18].

Having determined that the BioTAC system could identify BRD4 as a (+)-JQ1 target in focused screens, we sought to evaluate it as an unbiased target-ID method. To identify direct binders of (+)-JQ1 proteome-wide, we performed BioTAC proximity labeling experiments as described above, followed by label-free mass spectrometry-based proteomic analysis using data-dependent acquisition (DDA), of biotinylated proteins enriched following a 30 min treatment with 100 μM biotin plus DMSO or 1 μM Cpd 1, and then 1 μM Cpd 1 off-competed by pre-treatment with 10 μM free (+)-JQ1. We observed highly selective enrichment of known (+)-JQ1 targets BRD3 and BRD4, comparable to published (+)-JQ1 μMapping and superior to published (+)-JQ1 photo-affinity labeling (Fig. 1D, Supplementary Fig. 3A–D, Supplementary Fig. 4, Supplementary Data 1).

Having established conditions for determining the primary targets of small molecules using the BioTAC system, we next investigated its utility in reading out the interactome sphere of (+)-JQ1 bound BET-proteins. An advantage of the BioTAC system is the relatively long half-life of the activated biotin-AMP intermediate generated by the miniTurbo enzyme, allowing the labeling radius to be increased up to 35 nm by extending the labeling time[14]. To quantitively evaluate the ability of the BioTAC system to successfully enrich the known interactome of (+)-JQ1 bound BET proteins, we performed time course BioTAC proximity labeling experiments at the 1 h and 4 h time points, and evaluated streptavidin-enriched biotinylated proteins using mass spectrometry-based proteomic analysis (Fig. 2A, Supplementary Fig. 4, Supplementary Data 1). As expected, longer time points correlated with a greater number of proteins meeting our significance cut-offs (defined as FC > 2, P < 0.05) for both enrichment in Cpd 1 vs. DMSO, and competition with free (+)-JQ1 in Cpd 1 vs. Cpd 1 + 10 μM (+)-JQ1. Known (+)-JQ1 direct targets BRD3 and BRD4 were significantly enriched and off-competed at all time points and BRD2 was also detected

at the 4 h timepoint. Encouragingly, identified hit proteins at 1–4 h also included many known BET interactors, such as polymerase associated factor PAF1 (Supplementary Data 1)[19].

To evaluate the robustness of our technique across different proteomic instruments and data acquisition strategies, we repeated the BioTAC proximity labeling experiments and evaluated streptavidin enriched proteins with an alternative mass spectrometer type (Time of Flight, previous Orbitrap) at the 30 min time point, using a data-independent acquisition (DIA) method (Fig. 2B). Again, we successfully identified known (+)-JQ1 binders BRD2, BRD3, and BRD4, as well as known BRD2/3/4 interactors, including ATAD5, KMT5B, TAF1 and components of the mediator complex[19]. Finally, we identified importin 9 (IPO9), a nuclear import receptor that recognizes nuclear localization signals[20], such as those on BRD4, indicating that IPO9 may be one of the proteins responsible for the active transport of BRD4-complexed miniTurboFKBP12[F36V] into the nucleus. Together, these data indicate that the BioTAC method is a robust and widely applicable technique compatible with multiple experimental formats.

The detection of chromatin-associated proteins in the Cpd 1 BioTAC proteomic experiments suggested that we were labeling direct interactors of (+)-JQ1 bound BRD2, 3, and 4. To test this hypothesis we compared our compiled hits from both DDA and DIA experiments at all time points to an extensive BET protein interactome reference dataset identified using AP/MS and PL/MS with and without (+)-JQ1, reported by Lambert et al.[21]. The BioTAC system afforded a candidate interactor list significantly enriched in complexed proteins identified by Lambert et al., without requiring a priori knowledge of direct (+)-JQ1 binders[21]. We were pleased to discover that 41.7% of our statistically significant enriched and (+)-JQ1 rescued hits were also present in the Lambert dataset. This observation falls greater than 88 standard deviations above a random bootstrap analysis of the human proteome (Fig. 2C). To account for the limitations of using the findings of only one study as a benchmark, and our observation that many non-overlapping proteins are chromatin-associated, we performed Gene Ontology (GO) Biological Process analysis of the hits identified by BioTAC[22,23]. Here, transcriptional elongation by Pol II was significantly enriched (P < 0.05), consistent with the known functions of BET protein containing complexes targeted by bromodomain inhibitors (Fig. 2D)[24].

The linker conjugation site and exit vector on a compound-of-interest may affect the bound interactome due to both steric effects on direct target binding, and allosteric effects, as derivatized binders may affect their target proteins conformation and therefore interact differently. Therefore, we sought to characterize how altering the linker conjugation site of (+)-JQ1 in BioTAC molecules would alter the enriched interactome. Alternative (+)-JQ1 linker conjugation sites have been described by Hsia et al., who showed that functionalization at the 4-phenyl position with a linker conjugating to an aryl sulfonamide-based BRD4 binding ligand promoted intramolecular BRD4 binding, resulting in increased complexation with DCAF16, ultimately leading to BRD4 degradation[25]. Importantly, Hsia et al. also showed that 4-phenyl functionalization alone does not promote BRD4 degradation, indicating that this may be an optimal site for our approach[25]. We synthesized bifunctional (+)-JQ1 derivatives conjugated at the 4-phenyl position to orthoAP1867 (Fig. 3A, Supplementary Fig. 5A), and confirmed bifunctional compound-dependent enrichment of BRD4, and rescue of BRD4 labeling in the presence of excess (+)-JQ1 in BioTAC experiments at the 30 min time point by immunoblot (Fig. 3B, Supplementary Fig. 5B). Next, we performed BioTAC proximity labeling experiments at the 4 hr time point with Cpd 2, and evaluated streptavidin-enriched biotinylated proteins using mass spectrometry-based proteomic analysis (Fig. 3C–E). We successfully identified BRD2, BRD3 and BRD4 as interactors of 4-phenyl functionalized (+)-JQ1 derivatives, along with a number of known BRD2/3/4 interacting proteins, and components of the transcriptional machinery (Fig. 3D, E). To evaluate how the identified (+)-JQ1-interacting proteins were affected

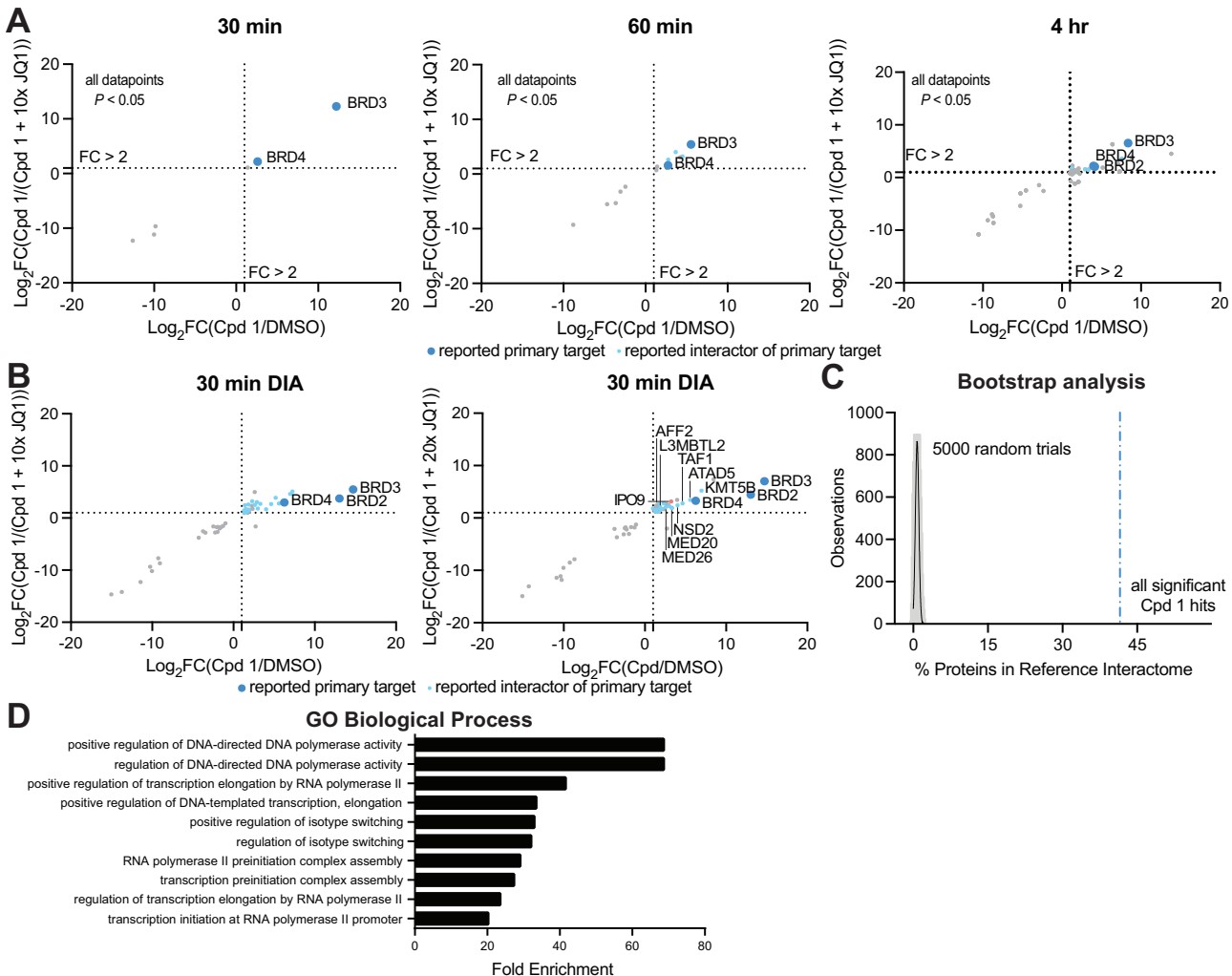

**Fig. 2 | The BioTAC system enables rapid, accurate small molecule interactome-ID. A** Time-course proteomics using data-dependent acquisition methods of streptavidin-enriched biotinylated proteins isolated from HEK293 cells transiently transfected with miniTurboFKBP12[F36V] and treated with the indicated compounds and 100 µM biotin, demonstrating enrichment and competition of known direct targets (30 min) and complexed proteins (60 min, 4 h). Only proteins with P-value < 0.05 (two-sample moderated T-test) in both conditions depicted. FC = fold-change, complete datasets in Supplementary Data 1, plotted individually in Supplementary Fig. 4. High-confidence hits are defined as those that are enriched >2-fold in both Cpd **1**/ DMSO and Cpd **1**/Cpd **1** + 10x (+)-JQ1, where P < 0.05, plotted upper-right quadrant. **B** Proteomics using data-independent acquisition methods of streptavidin-enriched biotinylated proteins isolated from HEK293 cells transiently transfected with miniTurboFKBP12[F36V] and treated with the indicated compounds and 100 µM biotin for 30 min, demonstrating enrichment and competition of known direct targets and complexed proteins. Only proteins with P-value < 0.05 (two-sample moderated T test) in both conditions depicted. FC = fold-change, complete datasets in Supplementary Data 1. **C** Percent enrichment of reference interactors (Lambert et al.[21]) in the hits identified from datasets depicted in panels **A** and **B** vs. the percent enrichment of reference interactors in 5000 random protein sets of equivalent size from the human transcriptome, showing significant enrichment by Cpd **1** (>41 standard deviations away from mean of random chance). **D** Gene Ontology analysis, showing significant enrichment of biological processes associated with known BET-protein function in the datasets depicted in panels **A** and **B**. Source data are provided as a Source Data file.

by the linker-attachment site, we compared the hit lists from BioTAC experiments performed using either Cpd **1** or Cpd **2** (Fig. 3C, data from Figs. 2B and 3D, E). We observed that a core set of 8 interactors were identified in both experiments, that included direct targets BRD2, BRD3 and BRD4, as well as known BRD interactors ATAD5, KMT5B, NSD3 and UBTF. In addition, GSTK1, a glutathione transferase associated with functions in cellular detoxification and small molecule metabolism, indicating that Cpd **1** and Cpd **2** may be substrates of GSTK1 in cells. In addition to these overlapping hits, a substantial number of chromatin-associated proteins were identified by Cpd **2**, but not Cpd **1** and vice versa. These results confirm that different linker conjugation sites allow access to, and labeling of, different ligand-bound target complexation states.

To further confirm that the BioTAC system can identify the cellular targets of small molecules, we extended our analysis to a second model

ligand, Alisertib. Alisertib is a highly selective Aurora kinase A inhibitor, and we and others have reported linker attachment sites in the context of targeted protein degraders that permit Aurora A recruitment[26,27]. We synthesized a panel of bifunctional molecules linking Alisertib to orthoAP1867 (Fig. 4A, Supplementary Fig. 6A), and confirmed Aurora A enrichment by immunoblot in BioTAC experiments in the presence of Cpd **3**, and dose-dependent competitive rescue in the presence of excess Alisertib. We selected the molecules with the best cell permeability (Cpd **3**, Supplementary Fig. 6C), and performed BioTAC proximity labeling experiments at the 4 h time point with Cpd **3**, and evaluated streptavidin-enriched biotinylated proteins using mass spectrometry-based proteomic analysis, where we successfully identified Aurora A as the major enriched target (Fig. 4C, D). These results confirmed that the BioTAC System is a reliable method for live cell target-ID across numerous molecules and targets.

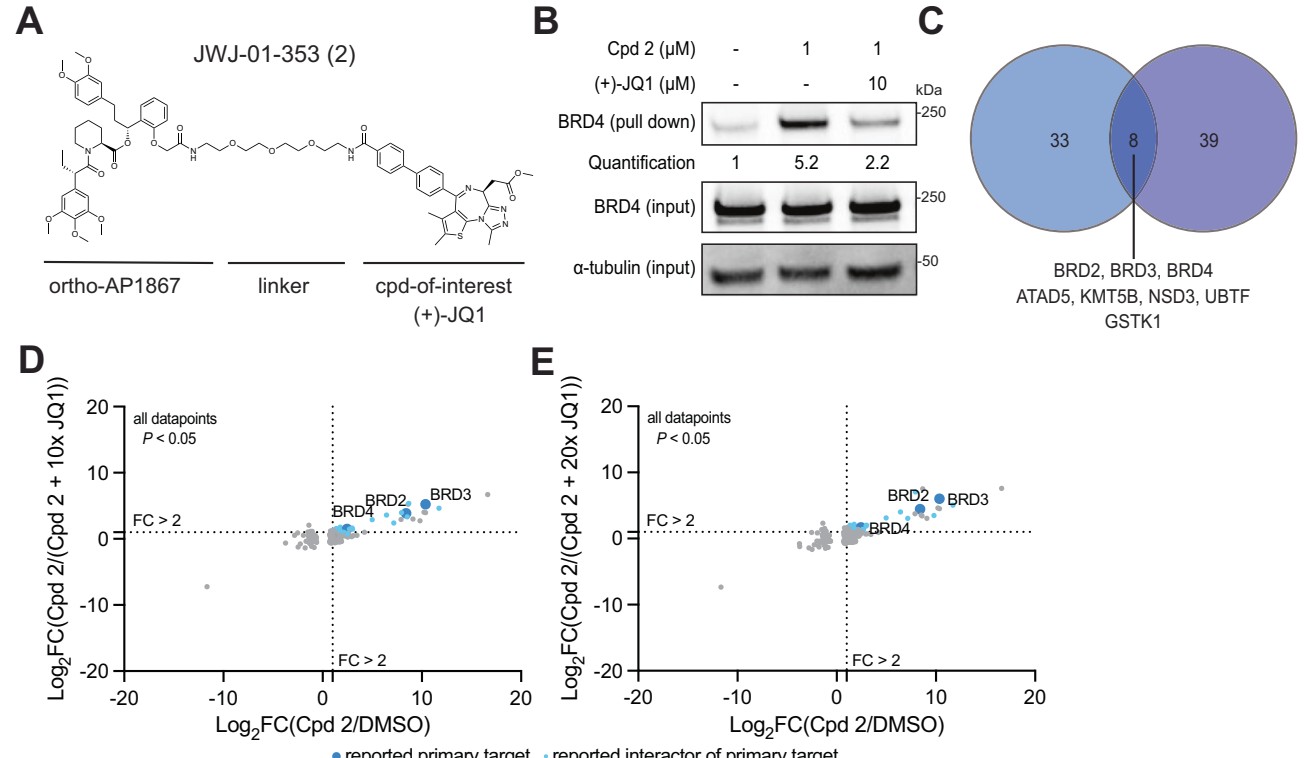

**Fig. 3 | Linker Exit Vector Diversification Identifies Core and Extended Interactors. A** Chemical structure of (+)-JQ1 bifunctional with alternative linker attachment site. **B** Immunoblot analysis of BRD4 enrichment following treatment of HEK293 cells transiently transfected with miniTurboFKBP12$^{F36V}$ with the indicated compounds and 100 μM biotin at the 30 min timepoint (input = sample processing control, $n = 2$ biological replicates, Supplementary Fig. 5B). **C** Venn diagram showing statistically significant hits from DIA BioTAC experiments at the 30 min time point using Cpd **1** (blue) or at the 4 h time point from Cpd **2** (purple).

Overlapping hits are listed below. **D**, **E** Proteomics using data-independent acquisition methods of streptavidin-enriched biotinylated proteins isolated from HEK293 cells transiently transfected with miniTurboFKBP12$^{F36V}$ and treated with DMSO, 1 μM Cpd **2**, plus the indicated compounds and 100 μM biotin for 4 h, demonstrating enrichment and competition of known direct targets and complexed proteins. Only proteins with $P$-value < 0.05 (two-sample moderated T test) in both conditions depicted. FC fold-change, complete datasets in Supplementary Data 1. Source data are provided as a Source Data file.

The paucity of practical methods for rapid, unbiased readout of small-molecule induced interactome changes has hindered the rational discovery and development of molecular glues. Molecular glues are small molecules which exert their function by to binding a primary target, and promoting or strengthening its interaction with a second protein through interactions at the protein-protein interface[28]. Molecular glue discovery is an area of high biomedical interest due to the ability of molecular glues to target undruggable oncoproteins such as transcription factors, which are recalcitrant to traditional inhibitor discovery but can be neutralized by induced complexation and targeted degradation[28]. However, as the binary affinity between a molecular glue and the second recruited target is low or non-existent in the absence of the primary target, molecular glue interactions are challenging to detect and screen for. Having rigorously benchmarked the performance of the BioTAC system using well-characterized (+)-JQ1 and Alisertib, we sought to use the BioTAC system to inform on complexes assembled by molecular glues (Fig. 5A)[28]. We selected trametinib as a non-degrader glue with which to benchmark the platform. Trametinib derives its clinical anti-cancer efficacy from promoting the interaction of its primary targets MEK1/2 with KSR1/2, but has low affinity for KSR1 or KSR2 alone, meaning this interaction was missed during clinical development[1].

We synthesized bifunctional trametinib analog Cpd **4**, with the linker attachment site informed by a reported trametinib-derived BODIPY-linked BRET probe named Tram-bo (Fig. 5B)[1]. We evaluated the cell permeability of Cpd **4** as described above, which was less cell permeable than the (+)-JQ1 derivatives (Supplementary Fig. 7A). Nevertheless, Cpd **4** supported efficient MEK1 labeling, following dose

and time point optimization (Supplementary Fig. 7B, C). We evaluated the ability of the BioTAC system to detect both known interactors of trametinib by Western blot, as described above. We successfully detected the MEK1:trametinib:KSR1 complex using the BioTAC system following a 4 h treatment with 1 μM Cpd **4**, and dose-dependent competition in the presence of trametinib of both KSR1 and MEK1 by immunoblot (Fig. 5C, Supplementary Fig. 7C). Low expression of KSR1 in HEK293 cells resulted in lower KSR1 signal relative to MEK1 in both input and enriched samples, indicating that a small proportion of MEK1 in HEK293s is bound by KSR1 in the presence of trametinib. In line with findings by Khan et al. that trametinib binds more tightly, and with a slower off-rate ($K_{dis}$) to MEK1-KSR1 than to MEK1 alone, higher concentrations of trametinib were required to off-compete labeling of KSR1, though signal-to-noise was low[1]. To strengthen our confidence that the MEK1: trametinib: KSR1 complex can be reliably detected by BioTAC, and to allow quantification of enrichment and competition, we turned to published conditions from Khan et al.[1]. We repeated the BioTAC experiment, this time with full-length mouse KSR1 (mKSR1) overexpressed concomitantly with miniTurbo-FKBP12$^{F36V}$, and observed significant enrichment of KSR1 in the presence of Cpd **4** and convincing off-competition by trametinib (Fig. 5D, Supplementary Fig. 7D, E). We next performed BioTAC proximity labeling experiments at the 4 h time point with Cpd **4**, and evaluated streptavidin-enriched biotinylated proteins using mass spectrometry-based proteomic analysis, where we successfully identified MEK1 (MAP2K1) and MEK2 (MAP2K2) as the major enriched targets, as well as KSR1, MEK1/2 associated proteins ARAF and BRAF, and putative off-target MEK5 (MAP2K5) (Fig. 5E).

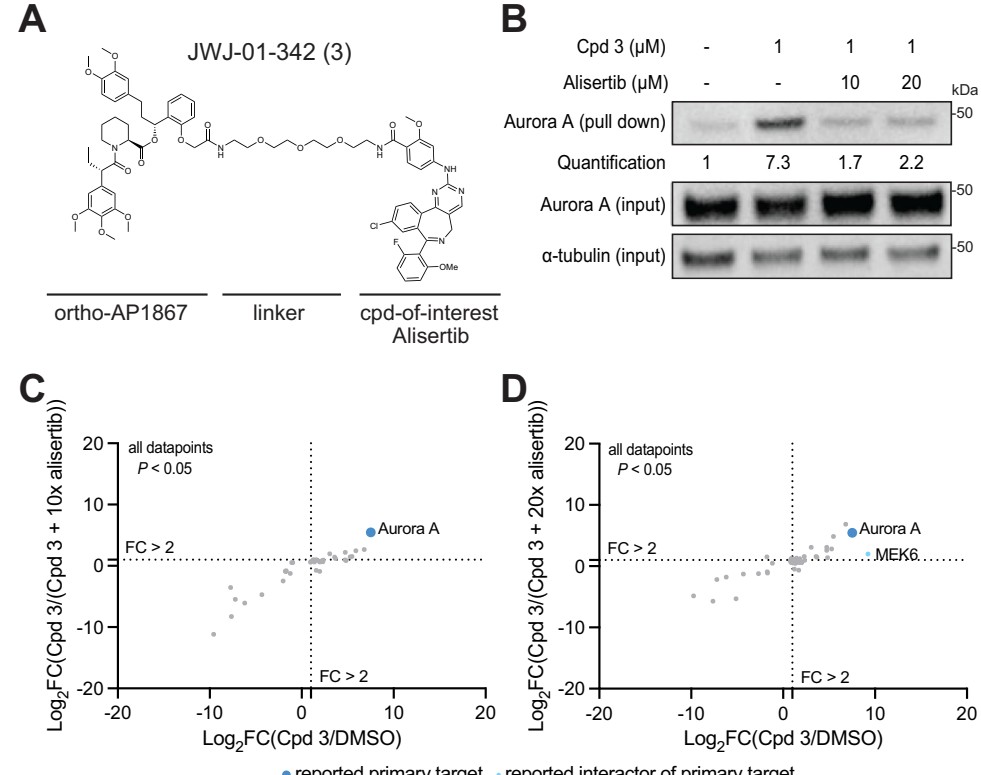

**Fig. 4 | The BioTAC system is an extensible strategy for live-cell target-ID.**
**A** Chemical Structure of Cpd **3**. **B** Immunoblot analysis of Aurora A enrichment following treatment of HEK293 cells transiently transfected with miniTurboFKBP12$^{F36V}$ with the indicated compounds and 100 μM biotin at the 4 h timepoint (input = sample processing control, $n$ = 2 biological replicates).
**C, D** Proteomics using data-independent acquisition methods of streptavidin-enriched biotinylated proteins isolated from HEK293 cells transiently transfected

with miniTurboFKBP12$^{F36V}$ and treated with DMSO, 1 μM Cpd **3**, plus the indicated compounds and 100 μM biotin for 4 h, demonstrating enrichment and competition of known direct targets and complexed proteins. Only proteins with $P$-value < 0.05 (two-sample moderated T test) in both conditions depicted. FC fold-change, complete datasets in Supplementary Data 1. Source data are provided as a Source Data file.

To extend our investigation of MEK-binding molecular glues, we next prepared derivatives of trametiglue, a trametinib derivative designed to enhance the association of MEK1/2 and RAF proteins[1]. Using the same linker-attachment site, we prepared Trametiglue-linked bifunctional molecules (Fig. 5F, Supplementary Fig. 7F), and demonstrated their ability to induce MEK1/2 biotinylation in BioTAC experiments by immunoblot at the 4 h time point, selecting Cpd **5** for further characterization (Supplementary Fig. 7G, H). Next, we performed BioTAC proximity labeling experiments at the 4 h time point with Cpd **5**, and evaluated streptavidin-enriched biotinylated proteins using mass spectrometry-based proteomic analysis, where we again successfully identified MEK1 (MAP2K1) and MEK2 (MAP2K2) as the major enriched targets, as well as KSR1 and ARAF, which demonstrated a modest enhancement in fold-change compared to that observed with Cpd **4** (Fig. 5G).

To confirm that labeling was occurring in the biologically relevant compartment (the cytosol), we imaged cells transfected with HA-tagged miniTurbo-FKBP12$^{F36V}$, and treated with DMSO, 1 μM Cpd **4**, or 1 μM Cpd **5**. In all conditions, miniTurbo-FKBP12$^{F36V}$ was exclusively localized to the cytosol (Supplementary Fig. 8). Together, these data demonstrate the ability of the BioTAC system to detect the complexes assembled by non-degrader molecular glues.

## Discussion

Methods for the routine measurement of drug-target interactomes are lacking, hiding the mechanism of action of numerous small molecules from view. Even though interactome remodeling in response to small molecule drugs is a common phenomenon that mediates drug efficacy

and resistance, little progress has been made in identifying and characterizing such events. Here we report the BioTAC system, which can identify the direct target of a small molecule, as well as its complexed proteins with high confidence in cytosolic and nuclear compartments. We demonstrate successful enrichment of the reported interactome of the epigenetic inhibitor (+)-JQ1, and measure the live cell interactome of 3 additional ligands, but this approach is theoretically applicable to any small molecule of interest that can be functionalized with a linker. Once caveat, is that the BioTAC system has not been evaluated for its accuracy in labeling targets that reside inside other membrane enclosed organelles.

The BioTAC system uses a universal recruitable biotin ligase chimera (miniTurboFKBP12$^{F36V}$) expression construct, facilitating rapid application to any system of interest without the need for cloning or cell line modification, rather implementing a stable cell line approach (as opposed to the transient transfection approach described here), in the absence of an inducible promoter produces background labeling incompatible with the method. Bifunctional molecules for investigating any drug-of-interest are synthesized in one step from a common ortho-AP1867 precursor using robust coupling chemistries. We recommend use of FKBP12$^{F36V}$ cellular target engagement assays to characterize cell permeability of bifunctional molecular, and if the phenotype or a target is already known, confirmation of the activity of linker derivatized molecules can aid candidate selection. We believe high construct expression relative to bifunctional compound cellular concentration is desirable, and therefore, compounds with reduced cell permeability are well tolerated in this system. Finally, the enrichment, mass spectrometry and data analysis methods are adapted from

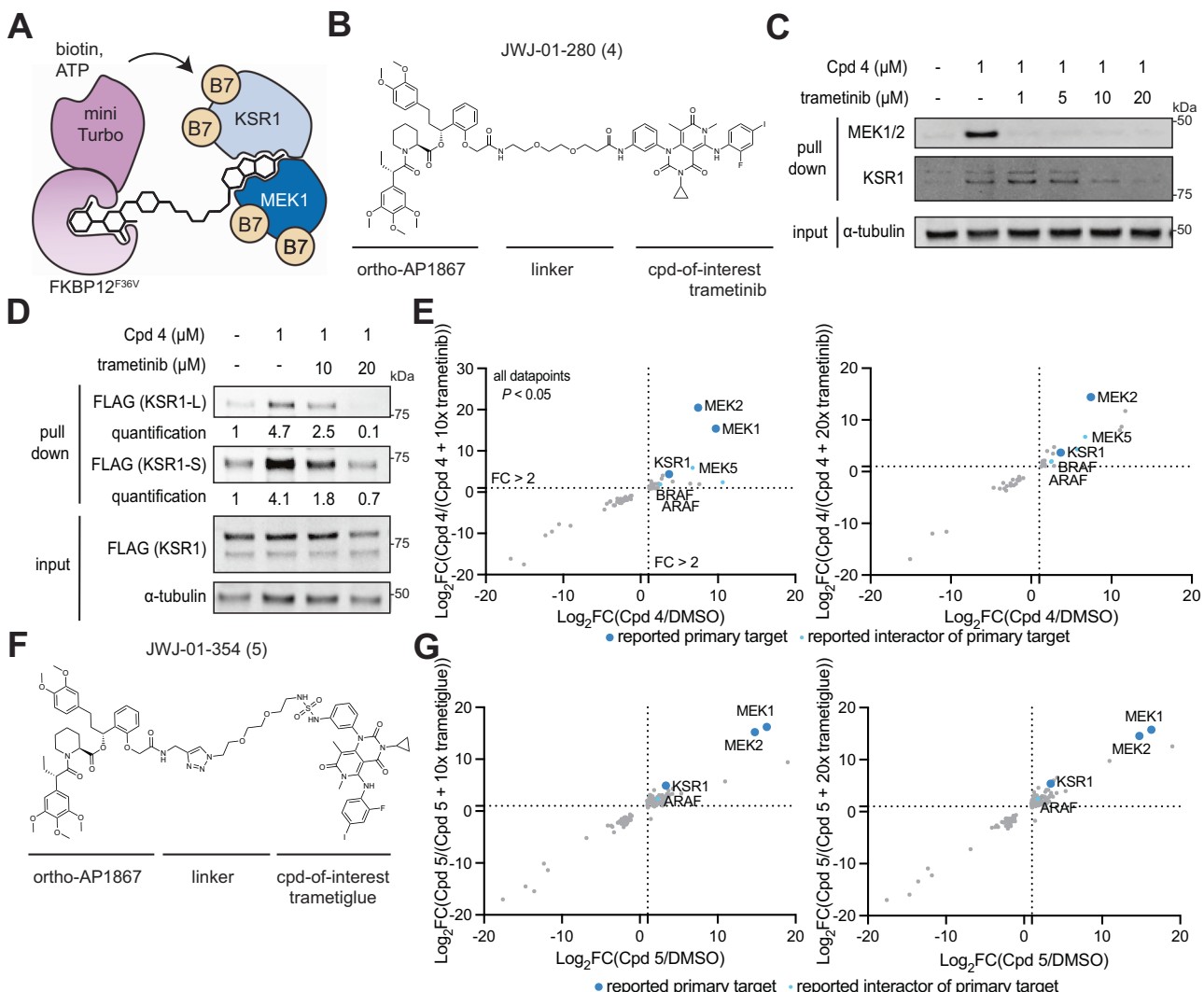

**Fig. 5 | The BioTAC system enables detection of non-degrader molecular glue interactions. A** Schematic depicting how the BioTAC system can detect molecular glue interactions. **B** Chemical structure of trametinib bifunctional molecule Cpd **4**. **C** Immunoblot analysis of MEK1 and KSR1 enrichment following treatment of HEK293 cells transiently transfected with miniTurboFKBP12[F36V] with the indicated compounds and 100 μM biotin at the 4 h timepoint, and streptavidin-based enrichment, showing successful enrichment and competition with trametinib (input = sample processing control, n = 2 biological replicates). **D** Immunoblot analysis of mKSR1 enrichment following treatment of HEK293 cells transiently transfected with miniTurbo-FKBP12[F36V] and mKSR1, with the indicated compounds and 100 μM biotin at the 4 h timepoint, and streptavidin-based enrichment, showing successful enrichment and competition with trametinib (input = sample processing control, n = 3 biological replicates, see Supplementary Fig. 7D, E) In **C** and **D** two KSR1 isoforms are observed, produced by alternative splicing[34]. KSR1-L (102 KDa) corresponds to the expected MW of Uniprot Q8IVT5-1 (canonical sequence), KSR1-S (87 KDa) corresponds to the expected MW of variant with residues 1–137 missing, Uniprot Q8IVT5-3, and −4. Our data indicate both isoforms can

complex with trametinib-bound MEK1. **E** Proteomics using data-independent acquisition methods of streptavidin-enriched biotinylated proteins isolated from HEK293 cells transiently transfected with miniTurboFKBP12[F36V] and treated with either DMSO or 1 μM Cpd **4** plus the indicated compounds and 100 μM biotin for 4 h, demonstrating enrichment and competition of known direct targets and complexed proteins. Only proteins with P-value < 0.05 (two-sample moderated T test) in both conditions depicted. FC = fold-change, complete datasets in Supplementary Data 1. **F** Chemical structure of trametiglue bifunctional molecule Cpd **5**. **G** Proteomics using data-independent acquisition methods of streptavidin-enriched biotinylated proteins isolated from HEK293 cells transiently transfected with miniTurboFKBP12[F36V] and treated with either DMSO or 1 μM Cpd **5**, plus the indicated compounds and 100 μM biotin for 4 h, demonstrating enrichment and competition of known direct targets and complexed proteins. Only proteins with P-value < 0.05 (two-sample moderated T test) in both conditions depicted. FC fold-change, complete datasets in Supplementary Data 1. Source data are provided as a Source Data file.

standard protocols in proximity labeling already performed by most proteomics core facilities, and have been tested using multiple instruments and data acquisition strategies[13]. These features make the BioTAC system readily accessible for broad application. During the preparation of this manuscript, a preprint describing a related approach for identifying targets of small molecules via SNAP- and Halo-tagging of TurboID was also disclosed[29]. Whilst these systems differ from the BioTAC system in their reported specificity and have not yet been benchmarked for interactome-detection, their successful

implementation across a range of ligands highlights the robustness of using proximity labeling to interrogate small molecule targets[29].

Next, we show the BioTAC system can identify molecular glue pairs that previously evaded detection, in a single experiment using Trametinib as a model system. The discovery and detection of molecular glue interactions is challenging to evaluate, in particular when evaluating non-degrader molecular glues. Recently, elegant methods to identify non-degrader molecular glues have been described, where size exclusion chromatography coupled to mass spectrometry is used

in combination with activity-based protein profiling to screen electrophilic compound libraries, yielding covalent stabilizers and disruptors of protein-protein interactions[30]. However, these approaches require resource intensive workflows, and are limited to covalent glues compounds, highlighting a need for additional approaches. In future applications, we envision the BioTAC system may find utility as a complementary screening method, for unbiased profiling of putative glue libraries. It is important to note, however, that the BioTAC system coupled to mass spectrometry, does not differentiate between proteins assembled through a glue interaction, and proteins that complex ligand bound targets, therefore all glue candidates should be validated using orthogonal biochemical assays.

In the long term, we anticipate that building community-wide knowledge around how small molecule drugs alter their target proteins complexation will lay the foundation for the rational design of drug target interactome profiles, to combat drug resistance, and enable wider targeting of the undruggable proteome.

# Methods

## Cell culture
The following cell lines were employed in this study: HEK293 (ATCC, CRL-1573), A549 (ATCC, CCL-185), HCT116 (ATCC, CCL-247), HEK293FT-FKBP12$^{F36V}$Nluc (N. S. Gray lab). Cells were cultured in high-glucose DMEM (Gibco, 11965092) with 10% FBS (Gibco, 10437028) and 1% Penicillin–Streptomycin (Gibco, 15140122). All cell lines were maintained in 37 °C and 5% $CO_2$ incubators and routinely tested negative for mycoplasma contamination using the MycoAlert Kit (Lonza, LT07318).

## Plasmids
*pCDNA3.1(+) miniTurboID-6xGGSG linker-FKBPF36V-2xHA* − DNA encoding the BioTAC construct (see Supplementary Data 1) was synthesized at GenScript and cloned into the MCS of pCDNA3.1(+) by NEBuilder HiFi DNA Assembly (New England Biolabs, E2621).

*pCMV5 WT KSR1* − Full-length mouse KSR1-FLAG was obtained from Addgene (25970).

*pCMV5 W781D KSR1* − KSR1(W781D) mutant was created from pCMV5 WT KSR1 using PrimeSTAR Max (Takara, R045A) mutagenesis PCR with forward (caatctgGTCgatcaaggcctcagcaggctgg) and reverse (cttgatcGACcagattggaagtggggaaggagtacg) primers according to the manufacturer's instructions.

## Immunoblotting
0.6 million HEK293 cells were plated into wells (Corning, 3506) containing 1 mL of media and incubated overnight. Cells were transfected with 100 μL of OptiMEM (Gibco, 11058021) containing 1 μg of BioTAC plasmid and 3 μL of TransIT-2020 transfection reagent (Mirus, MIR5400) for 24 hr. For KSR1-FLAG co-transfections 1 μg of KSR1-FLAG plasmid was mixed with BioTAC plasmid in OptiMEM prior to adding transfection reagent. Following a media change, cells were pre-treated with parental molecules at indicated concentrations or vehicle (up to 1% DMSO) for 10 min, then treated with bifunctional compound at indicated concentrations or vehicle (0.1% DMSO) for 10 min, and finally labeled with 100 μM biotin for indicated times. Cells were washed twice with DPBS (Gibco, 14190144) then directly lysed with 200 μL RIPA lysis buffer (50 mM Tris pH 8, 150 mM NaCl, 0.1% SDS, 0.5% sodium deoxycholate, 1% Triton X-100) containing fresh 1x HALT inhibitor cocktail (Thermo, 78442), 100 μM PMSF, and 125 U/mL benzonase (MilliporeSigma, 706643), transferred to 1.5 mL tubes, and chilled on ice for 15 min. Lysates were clarified by centrifuging at 16,000 x g for 10 min at 4 °C, then protein concentration was determined by BCA assay (Thermo, 23225). Input samples were prepared in 1x NuPage LDS sample buffer (Invitrogen, NP0007) and boiled at 95 °C for 10 min. 120-240 μg of protein was normalized to 500 μL in RIPA lysis buffer and rotated overnight at 4 °C with a 12:1 protein-to-bead (w/v) ratio of

streptavidin magnetic beads (Thermo, 88817) that was pre-washed with RIPA lysis buffer. Beads were washed and eluted as described in Cho et al. [13]. Briefly, beads were washed twice with RIPA lysis buffer, once with 1 M KCl, once quickly with 0.1 M sodium carbonate, once quickly with 2 M urea in 50 mM Tris pH 8, and twice with RIPA lysis buffer. Following the final wash, beads were boiled in 1x NuPage LDS sample buffer containing 2 mM biotin and 20 mM DTT at 95 °C for 10 min.

SDS-PAGE samples were run on a 4–12% bis-tris precast gel (Invitrogen, NW04125BOX) in MES buffer (Invitrogen, B0002) for 45 min at 180 V, then transferred to a nitrocellulose membrane (Bio-Rad, 1620112) in Bolt transfer buffer (Invitrogen, BT00061) for 90 min at 45 V. Membrane was blocked with 5% non-fat dry milk (Kroger) in TBST (Thermo, 28360) for 1 h then incubated with primary antibody diluted in TBS blocking buffer (LI-COR, 92760001) overnight at 4 °C. Primary antibodies used were 1:2000 BRD4 (Bethyl Laboratories, A301-985A-M), 1:2000 MEK1/2 (Cell Signaling Technology, 9122S), 1:2000 KSR1 (Cell Signaling Technology, 4640S), 1:2000 Aurora A (Cell Signaling Technology, 14475T), 1:2000 HA-tag (Cell Signaling Technology, 3724S), 1:5000 DYKDDDDK Tag (Cell Signaling Technology, 14793S), and 1:2000 alpha-tubulin (Cell Signaling Technology, 3873S) as loading control. Membrane was washed three times with TBST then incubated with DyLight 680 anti-mouse IgG (Cell Signaling Technology, 5470S) and DyLight 800 anti-rabbit IgG (Cell Signaling Technology, 5151S) diluted 1:10000 in TBS blocking buffer for 1 h at room temperature. Membrane was washed three times with TBST then imaged with a ChemiDoc Imaging System (Bio-Rad) and quantified in Image Lab 6.1.0 (Bio-Rad).

Immunoblotting was additionally used to identify optimal cell line (Supplementary Fig. 9a), optimize transfection conditions with TransIT-2020 or Lipofectamine 3000 (Thermo, L3000015) (Supplementary Fig. 9b), and confirm biotinylation by probing membrane after blocking with 1:3000 IRDye 800CW streptavidin (LI-COR, 92632230) in TBS blocking buffer for 1 hr at room temperature then washing three times with TBST (Supplementary Fig. 9c).

## Dual-luciferase target engagement assay
Assay protocol was adapted from Nabet et al.[31]. using the Nano-Glo Dual-Luciferase Reporter Assay (Promega, N1630). Briefly, HEK293FT-FKBP12$^{F36V}$Nluc cells were plated at 4000 cells per well in 20 μL of media in a 384-well white microplate. Cells were incubated overnight then co-treated with 100 nM dTAG-13 and bifunctional compound at indicated concentrations in DMSO using a Pico 8 Digital Dispenser (Thermo). Cells were incubated for 5 hours at 37 °C and brought to room temperature before reagent addition. Cells were treated with 20 μL of ONE-Glo EX reagent and incubated on an orbital shaker for 10 min at RT, then Fluc luminesce was measured using a ClarioSTAR Plus microplate reader (BMG Labtech). Subsequently, cells were treated with 20 μL of NanoDLR Stop and Glo reagent and incubated on an orbital shaker for 10 min at RT, then Nluc luminescence measured. Nluc values were Fluc normalized, then the Nluc/Fluc ratios were normalized by the respective DMSO ratios. Data was plotted in GraphPad Prism 10.0.0 (GraphPad Software) and EC50 calculated using a log(inhibitor) vs response – variable slope.

## Cellular thermal shift assay
0.25 million HEK293 cells were plated into wells (Corning, 3513) containing 1 mL media and incubated for 24 h. Cells were treated with 1 μM compound, 10 μM compound, or vehicle (0.1% DMSO) then returned to the incubator for 4 h. Cells were washed once with PBS then harvested by trypsinization (Gibco, 25200056). Pellets were washed once with PBS, resuspended in 100 μL PBS containing fresh 1x HALT inhibitor cocktail, and transferred to PCR tubes. Tubes were incubated for 3 min in a thermal cycler at 47.5 °C for BRD4 compounds, 62 °C for MEK1/2 compounds, or 51.5 °C for Aurora A compounds then

incubated for 3 min at 25 °C. Cells were freeze-thawed three times in LN2, vortexing after each thaw. Lysates were transferred to 1.5 mL tubes and clarified by centrifuging at 16,000 × g for 10 min at 4 °C. Protein concentration was determined by BCA assay and SDS-PAGE samples were prepared with normalized amounts of protein in 1x NuPage LDS sample buffer. Immunoblotting was performed as described in the Immunoblotting section.

## Confocal Microscopy

Coverslips (Ibidi, 80426) were treated with 12 µg/mL poly-L-lysine in sterile water (ScienCell Research Laboratories, 0413) and incubated overnight. Coverslips were washed twice with sterile water then immediately plated with 0.15 million HEK293 cells in 0.5 mL media and incubated overnight. Cells were transfected with 25 µL OptiMEM containing 0.25 µg of BioTAC plasmid and 0.75 µL of TransIT-2020 transfection reagent for 24 hr. Following a media change, cells were treated with 1 µM of bifunctional compound or vehicle (0.1% DMSO) for 10 min then labeled in 100 µM biotin for 30 min or 4 h. Cells were washed three times with PBS, fixed with 4% formaldehyde in PBS for 10 min, washed three times with PBS, permeabilized with 0.1% Triton X-100 in PBS for 10 min, washed three times with PBS, blocked with 5% normal horse serum (Vector Laboratories, S-2000) in PBS for 30 min at room temperature, and incubated with primary antibody diluted in blocking buffer overnight at 4 °C. Primary antibodies used were 1:1600 HA-tag (Cell Signaling Technology, 3724S), 1:1600 BRD4 (Cell Signaling Technology, 63759S), or 1:100 MEK1/2 (Cell Signaling Technology, 4694S). Cells were washed three times with PBS then incubated with 1:800 Alexa Fluor 488 AffiniPure Donkey Anti-Rabbit (Jackson ImmunoResearch, 711-545-152) and 1:800 Alexa Fluor 647 AffiniPure Donkey Anti-Mouse (Jackson ImmunoResearch, 715-605-151) diluted in blocking buffer for 1 h at room temperature in the dark. Cells were stained with 1 µg/mL DAPI (Thermo, 62248) in PBS for 5 min then washed three times with PBS and stored in PBS for imaging. Z-stacks were acquired on a Lecia SP8 confocal microscope with a 100x oil immersion objective controlled with LAS X software. Colocalization analysis was performed in FIJI 1.54f[32] on maximum Z projections using Colocalization Thresholds with constant intensity pixels.

## Quantitative proteomics (data-dependent acquisition)

9 million HEK293 cells were plated into 15 cm dishes (Corning, 430599) containing 12 mL of media and incubated overnight. Cells were transfected with 1 mL of OptiMEM containing 10 µg of BioTAC plasmid and 30 µL of TransIT-2020 transfection reagent for 24 h. Following a media change, cells were pre-treated with parental molecules at indicated concentrations or vehicle (1% DMSO) for 10 min, then treated with 1 µM of bifunctional compound or vehicle (0.1% DMSO) for 10 min, and finally labeled with 100 µM biotin for indicated times (n = 3 biologically independent samples per condition). Cells were washed 5 times with DPBS then harvested by scraping on ice into 1.5 mL low-binding tubes (Eppendorf, 0030108442). Cells were lysed with 1.2 mL RIPA lysis buffer (50 mM Tris pH 8, 150 mM NaCl, 0.1% SDS, 0.5% sodium deoxycholate, 1% Triton X-100) containing fresh 1x HALT inhibitor cocktail, 100 µM PMSF, and 125 U/mL benzonase, then sonicated with 15 ×1 second pulses at 40% power using a 5/64" probe (Qsonica) then chilled on ice for 15 min. Lysates were clarified by centrifuging at 16,000 × g for 10 min at 4 °C, then protein concentration was determined by BCA assay. 3 mg protein was normalized to 1 mL in RIPA lysis buffer and rotated overnight at 4 °C with 250 µL of streptavidin magnetic beads that was pre-washed twice in RIPA lysis buffer. Beads were washed twice with RIPA lysis buffer, once with 1 M KCl, once quickly with 0.1 M sodium carbonate, once quickly with 2 M urea in 50 mM Tris pH 8, twice with RIPA lysis buffer, and finally 4 times in 50 mM Tris pH 8. On the final wash, beads were transferred to clean tubes and frozen at −80 °C.

Beads were resuspended in 1x TNE buffer (50 mM Tris pH 8.0, 100 mM NaCl, 1 mM EDTA) containing 0.1% RapiGest SF (Waters, 186002122) then boiled at 95 °C for 5 min. Samples were reduced with 1 mM TCEP for 30 min at 37 °C, then carboxymethylated with 0.5 mg/mL of iodoacetamide for 30 min at 37 °C, and neutralized with 2 mM TCEP. Samples were digested with 5 µg trypsin (Thermo, 90058) overnight at 37 °C. RapiGest SF was degraded and removed by treating the samples with 250 mM HCl at 37 °C for 1 h followed by centrifugation at 16,000 × g for 30 min at 4 °C. The soluble fraction was then added to a new tube and the peptides were extracted and desalted using C18 desalting columns (Thermo Scientific, PI-87782). Peptides were quantified using BCA assay and a total of 1 µg of peptides were injected for LC-MS analysis.

Trypsin-digested peptides were analyzed by ultra high pressure liquid chromatography (UPLC) coupled with tandem mass spectroscopy (LC-MS/MS) using nano-spray ionization. The nanospray ionization experiments were performed using a Orbitrap fusion Lumos hybrid mass spectrometer (Thermo) interfaced with nano-scale reversed-phase UPLC (Thermo Dionex UltiMate™ 3000 RSLC nano System) using a 25 cm, 75-micron ID glass capillary packed with 1.7-µm C18 (130) BEHTM beads (Waters corporation). Peptides were eluted from the C18 column into the mass spectrometer using a linear gradient (5–80%) of ACN (Acetonitrile) at a flow rate of 375 µl/min for 1.5 h. The buffers used to create the ACN gradient were: Buffer A (98% H2O, 2% ACN, 0.1% formic acid) and Buffer B (100% ACN, 0.1% formic acid). Mass spectrometer parameters are as follows: an MS1 survey scan using the orbitrap detector, mass range (m/z): 400-1500 (using quadrupole isolation), 120000 resolution setting, spray voltage of 2200 V, ion transfer tube temperature of 275 °C, AGC target of 400000, and maximum injection time of 50 ms. MS1 survey scan was followed by data dependent scans using top speed for most intense ions, with charge state set to only include +2–5 ions, and 5 second exclusion time, while selecting ions with minimal intensities of 50000, in which the collision event was carried out in the high energy collision cell (HCD Collision Energy of 30%), and the fragment masses where analyzed in the ion trap mass analyzer (With ion trap scan rate of turbo, first mass m/z was 100, AGC Target 5000 and maximum injection time of 35 ms).

Protein identification was carried out in PEAKS Studio 10.6 (Bioinformatics Solutions) with the human reference proteome and the following parameters: semi-specific tryptic peptides, three missed cleavages, three variable modifications per peptide, 10 ppm parent mass error tolerance, 0.5 Da fragment mass error tolerance, and a 1% FDR cutoff. Missing values from identified proteins were imputed using the average of the lowest detected quantity in each sample. Quantities were DMSO normalized, then uploaded to Protigy 1.1.5 (Broad Institute) where data was log2 transformed and median-MAD scaled. A two-sample moderated T test and F test with nominal P-value filter of 0.05 was used to test for significant proteins.

## Quantitative proteomics (data-independent acquisition)

1.6 million HEK293 cells were plated into 6 cm dishes (Corning, 430196) containing 3 mL media and incubated overnight. Cells were transfected with 300 µL of OptiMEM containing 3 µg of BioTAC plasmid and 9 µL of TransIT-2020 transfection reagent for 24 h. Following a media change, cells were pre-treated with parental molecules at indicated concentrations or vehicle (0.2% DMSO) for 10 min, then treated with 1 µM of bifunctional compound or vehicle (0.1% DMSO) for 10 min, and finally labeled with 100 µM biotin for indicated times (n = 3 biologically independent samples per condition). Cells were washed 3 times with DPBS then directly lysed with 300 µL RIPA lysis buffer (50 mM Tris pH 8, 150 mM NaCl, 0.1% SDS, 0.5% sodium deoxycholate, 1% Triton X-100) containing fresh 1x HALT inhibitor cocktail, 100 µM PMSF, and 125 U/mL benzonase. Cells were transferred to 1.5 mL low-binding tubes and sonicated with one 5 second pulse at 40% power

using a 5/64" probe then chilled on ice for 15 min. Lysates were clarified by centrifuging at 16,000 x g for 10 min at 4 °C, then protein concentration was determined by BCA assay. 600 μg protein was normalized to 1 mL in RIPA lysis buffer and rotated overnight at 4 °C with 50 μL of streptavidin magnetic beads that was pre-washed with RIPA lysis buffer. Beads were washed twice with RIPA lysis buffer, once with 1 M KCl, once quickly with 0.1 M sodium carbonate, once quickly with 2 M urea in 50 mM Tris pH 8, twice with RIPA lysis buffer, and finally 3 times in 50 mM Tris pH 8. On the final wash, beads were transferred to clean tubes.

Beads were resuspended in 100 μL 50 mM ammonium bicarbonate containing 5 μl 5% RapiGest SF in 10x TNE buffer (500 mM Tris pH 8.0, 1 M NaCl, 10 mM EDTA) then boiled at 95 °C for 10 min. Samples were reduced and carboxymethylated in one step by adding 200 μL 50 mM ammonium bicarbonate containing fresh 3 μL 0.5 M TCEP and 9 μL 0.5 M chloroacetamide then shaking at 900 RPM on a Thermo-Mixer (Eppendorf) for 30 min at room temperature. Samples were digested with 1 μg trypsin at 900 RPM on a ThermoMixer overnight at 37 °C. RapiGest SF reagent was degraded and removed by adding 40 μL 1 M HCl (Sigma, H9892) and shaking at 900 RPM on a ThermoMixer at 37 °C for 30 min followed by centrifugation at 16,000 × g for 45 min at 4 °C. Beads were collected on a magnetic rack, then supernatants were carefully transferred to clean tubes and centrifuged at 16,000 × g for 30 min at 4 °C. Beads were collected on a magnetic rack and peptides were desalted with 0.6 μL C18 ZipTips (MilliporeSigma, ZTC18S096). Briefly, tips were equilibrated twice with 100% ACN then twice with 0.1% formic acid in water. Tips were loaded by aspirating and dispensing sample 10 times, then tips were washed three times with 0.1% formic acid in water. Peptides were eluted into clean tubes by aspirating and dispensing 5 times with 10 μL 0.1% formic acid in 80% ACN. Peptides were quantified by NanoDrop and 250 ng loaded onto Evotips (Evosep, EV2011) according to the manufacturer's instructions. Briefly, tips were equilibrated with 100 μL 0.1% formic acid in ACN, soaked in isopropanol for 60 seconds, and equilibrated with 100 μL 0.1% formic acid in water. Tips were loaded with peptides diluted in 100 μL 0.1% formic acid in water, then washed three times with 100 μL 0.1% formic acid in water. Tips were briefly spun down with 100 μL 0.1% formic acid in water and soaked in 0.1% formic acid in water at 4 °C until analysis.

The Evosep One LC system (Evosep) coupled with a timsTOF Pro 2 mass spectrometer (Bruker) was used to measure all samples. The 30 SPD (samples per day) methods required a 10 cm × 150 μm reverse-phase PepSep column packed with 1.5 μm C18 beads (Bruker) at 58 °C. The analytical columns were connected with a 10 μm ID fused silica emitter (Bruker) inside a CaptiveSpray nanoelectrospray ion source (Bruker). The mobile phases comprised 0.1% formic acid in water as solution A and 0.1% formic acid in ACN as solution B. The dia-PASEF method covered an MS1 m/z range from 100 to 1700 for proteome measurements. Each method included one IM window per dia-PASEF scan with variable isolation windows at 20 amu segments (table of PASEF windows in Supplementary Data 1). Mass step per cycle was set to 34. The accumulation and ramp times were specified as 100 ms for all experiments. The ion mobility (IM) settings were 0.85 for Start IM [1/K0], and 1.3 for End IM [1/K0]. As a result, each MS1 scan and each MS2/dia-PASEF scan lasted 100 ms plus additional transfer time, and dia-PASEF scans had a cycle time of 1.38 s. The collision energy was set to 10 ev.

Protein identification was carried out in DIA-NN 1.8.1[33] with an in silico spectral library generated from the human reference proteome (UP000005640_9606, version 2023-05-19). Default parameters, including 1% precursor FDR, were used except the following: canonical tryptic specificity (--cut K*,R*,!*P), two missed cleavages, two variable modifications per peptide, methionine oxidation and N-terminal acetylation included as variable modifications, 10 ppm mass accuracy, 10 ppm MS1 accuracy, and heuristic protein inference not selected. Missing values from proteotypic identifications

(unique_genes_matrix.tsv) were imputed using the average of the lowest detected quantity in each sample. Quantities were DMSO normalized, then uploaded to Protigy 1.1.5 (Broad Institute) where data was log2 transformed and median-MAD scaled. A two-sample moderated T test with nominal P-value filter of 0.05 was used to test for significant proteins.

## Reporting summary

Further information on research design is available in the Nature Portfolio Reporting Summary linked to this article.

## Data availability

The pCDNA3.1(+) miniTurboID-6xGGSG linker-FKBPF36V-2xHA plasmid has been deposited with Addgene under plasmid # 200641 [https://www.addgene.org/200641/]. The proteomic data generated in this study have been deposited in the PRIDE database under accession code PXD041401 and are available in Supplementary Data 1. Source data are provided with this paper.

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

## Acknowledgements

This work was supported by American Cancer Society IRG Grant # IRG-19-230-48-IRG, UC San Diego Moores Cancer Center, Specialized Cancer Center Support Grant NIH/NCI P30CA023100, NIH grant DP2NS132610 (F.M.F.), and NIH grant DP2GM146247 (J.G.E.). A.J.T. was supported by a UCSD Distinguished Graduate Student Award and the NIH/NCI Cancer Cell Signaling & Communication Training Grant T32 CA009523. G.E.G. was supported by the Molecular Biophysics Training Grant T32 32GM139795. P.G. was supported by an Undergraduate Summer Research Award. Proteomic studies were performed at the UC San Diego Biomolecular and Proteomics Mass Spectrometry Facility. Confocal microscopy was performed at the UC San Diego School of Medicine Microscopy Core (NINDS grant P30NS047101). The authors would like to thank Dr. Majid Ghassemian, Dr. Lisa Jones, Dr. Eric Bennett, Dr. Katherine A. Donovan and Dr. Xuezhen Ge for helpful discussions and feedback and Dr. Sonya Neal for use of a BioRad ChemiDoc Gel Imager.

## Author contributions

F.M.F. and J.G.E. conceived and led the study. J.G.E. and E.J.M designed and cloned the DNA constructs. J.W.J. and F.M.F. designed and performed molecule synthesis. A.J.T. and G.E.G. performed dual luciferase target engagement assays. A.J.T., P.G. and B.T.B. performed immunoblot experiments. A.J.T. and P.G. prepared samples for proteomic studies and thermal shift assays. A.J.T. performed confocal microscopy experiments. A.J.T., S.A.M. and J.G.E. performed proteomic data analysis. F.M.F. wrote the manuscript with input from all authors.

## Competing interests

A.J.T., J.J., J.G.E. and F.M.F. are inventors on a provisional patent application (63/394,145) relating to the invention of the BioTAC system and its use in characterizing small molecules jointly owned by the University of California San Diego, Dana-Farber Cancer Institute and University of Utah. F.M.F. is a scientific co-founder and equity holder in Proximity Therapeutics, an equity holder in Triana Biomedicines. Fleur Ferguson is or was recently a consultant or received speaking honoraria from Eli Lilly and Co., RA Capital, Tocris BioTechne, Janssen Pharmaceuticals, and Plexium Inc. The Ferguson lab receives or has received research funding from Ono Pharmaceutical Co. Ltd and Merck & Co. J.G.E. is a scientific co-founder, equity holder, and scientific advisory board (SAB) member in Evolution Bio. The English lab receives or has received research funding from Eli Lilly and Co. The remaining authors declare no competing interests.
