## [Peer Review File · Nature Communications]

REVIEWER COMMENTS

Reviewer #1 (Remarks to the Author):

This is a really great platform for rapidly identifying the direct and complexed small-molecule binding proteins of both inhibitors and molecular glues through their Biotin Targeting Chimera (BioTAC) platform. The approach employs heterobifunctional molecules linking the protein targeting ligand to an FKBP12(F36V) targeting ligand to be treated in cells expressing a FKBP12F36V fusion to miniTurbo to enable the biotinylation of the small-molecule target or molecular glue ternary complex proteins. They demonstrate proof-of-concept of their platform with JQ1 and identifying the bromodomain targets and trametinib to identify the molecular glue targets MEK1/2 and KSR1. This will be broadly applicable for target identification strategies and to identify protein complexes of future molecular glues. This study is well-done and rigorously performed and should be accepted as is.

Reviewer #2 (Remarks to the Author):

In this paper, the authors developed a tool to identify proteins surrounding a small molecule. Their tool requires two components, a chemical containing the small molecule conjugated to the FKBP12(F36V) binding compound ortho-AP1867 and miniTurbo fused to FKBP12(F36V). Introduction of the chemical leads to binding of both the small molecule's target and miniTurbo, enabling proximity labeling of its vicinity. The authors demonstrated their tool to study the interactors of (+)-JQ1 and trametinib. They performed western blotting for both to demonstrate enrichment of known interactors and proteomics for (+)-JQ1. They demonstrated off-competition of the interactors with addition of the unconjugated small molecule.

I recommend major revisions for the following reasons:

- The authors should address if their system alters the behavior of the small molecule itself and answer:
 1. How does conjugation to ortho-AP1867 affect permeability compared to the small molecule itself? Could it affect permeability not just through the plasma membrane but also other organelle membranes?
 2. Is the difference in permeability between cpd1 and cpd2 due to the difference in permeability of (+)-JQ1 and trametinib?
 3. Does the addition of the ortho-AP1867 handle and recruitment of miniTurbo affect the small molecule's behavior (ie increase steric hindrance that could block interactors)?
 4. How do the dose curves of the conjugated small molecule and small molecule itself compare?

- The authors should also clarify the cellular context of the tool and address the questions below:

5. In the cell, is the chemical in excess or is miniTurbo-FKBP12(F36V) in excess?

6. If miniTurbo-FKBP12(F36V) is in excess, is there noise from free, unassociated miniTurbo-FKBP12(F36V) that could biotinylate its surrounding? Does the expression of miniTurbo-FKBP12(F36V) have to be titrated to prevent excess, especially in cases where permeability of the chemical is limited? Could the authors provide streptavidin blots prior to enrichment?

7. Where does miniTurbo-FKBP12(F36V) localize in the cell? If it is in the cytosol, and the drug goes to the nucleus, will miniTurbo-FKBP12(F36V) binding impede the drug's ability to reach its target? Or perhaps miniTurbo-FKBP12(F36V) won't be efficiently loaded on to the chemical in the nucleus?

8. Imaging miniTurbo-FKBP12(F36V) by introducing an epitope tag would be nice and could potentially show differences in localization with and without chemical treatment and with different chemical treatments. Since cpd1 detects BRD4 which is annotated as nuclear and cpd2 detects KSR1 which is not annotated as nuclear, perhaps localization of miniTurbo-FKBP12(F36V) changes depending on whether it is treated with cpd1 or cpd2.

9. If the drug's target is in the mitochondria or secretory system, I suspect the tool would not work and you would need prior knowledge about the drug to target miniTurbo-FKBP12(F36V) to those organelles first.

- On the (+)-JQ1 section,

10. Does the drug bind to BRD2/3/4 one at a time or does BRD2/3/4 complex with each other?

11. If the drug binds to one BRD at a time, could the difference in enrichment be due to difference in affinities or difference in expression of BRD2/3/4?

- On the trametinib section,

12. The authors do not show anything to support its properties as a molecular glue. Their data does not show the differentiation in off-rate when MEK1 and KSR1 are both engaged versus MEK1 alone. The difference in the off-compete labeling of the MEK1 and KSR1 just demonstrates they bind with different affinities but not whether their affinities change depending on the presence of each other. Perhaps they should do knockout or knockdown of one of the components to show the difference in off-compete labeling when one of the interactors is absent.

- This tool could be more robustly validated and explored.

13. Western blotting of MS hits besides BRD4

14. Proteomics or blotting with Cpd2 samples to identify other proteins that complex around MEK1 and KSR1

15. Using BioTAC to deorphanize drug targets

- Additional minor points:

16. Increasing labelling time doesn't mean it increases labeling radius, but more so the coverage of labeling in the vicinity of the enzyme increases.

17. Double check the intro, uMap uses blue, not UV light.

Reviewer #3 (Remarks to the Author):

In this manuscript, Tao et al. developed the BioTAC system using bifunctional compounds and miniTurbo-fused FKBP12(F36V) to map small molecule interactomes. By using this BioTAC system, they detected compound-dependent biotinylation of direct and indirect interacting proteins of BET inhibitors. They also detected biotinylation of MEK1/2 and KSR1, the target proteins of the molecular glue trametinib.

However, it is surprising that the authors performed only focused screens by Western blotting to benchmark molecular glue-induced interactome changes. Can this BioTAC system detect compound-dependent significant changes in MEK1, MEK2, and KSR1 by label-free or TMT based quantitative mass spectrometry of biotinylated proteins? This unbiased, proteome-wide screening is essential to demonstrate that the BioTAC system is a comprehensive profiling method.

Minor comments:

1. TurboID and miniTurbo should be distinguished. Avoid using the terms miniTurboID or mTurbo.
2. The differences among Figure 1C, Figure S1I, and Figure S1J should be more clearly defined.
3. P-TEFb, TFIID, NuRD, and H4 should be plotted in Figure 2A.
4. Line 76, Figure S1d-e to Figure S1d-f.
5. Page 1 of Supporting Information, add "Supporting Figure 6 | Uncropped blots7"
6. Figure S6B, upper, positions of markers may be wrong.

REVIEWER COMMENT

Reviewer #1 (Remarks to the Author):

This is a really great platform for rapidly identifying the direct and complexed small-molecule binding proteins of both inhibitors and molecular glues through their Biotin Targeting Chimera (BioTAC) platform. The approach employs heterobifunctional molecules linking the protein targeting ligand to an FKBP12(F36V) targeting ligand to be treated in cells expressing a FKBP12F36V fusion to miniTurbo to enable the biotinylation of the small-molecule target or molecular glue ternary complex proteins. They demonstrate proof-of-concept of their platform with JQ1 and identifying the bromodomain targets and trametinib to identify the molecular glue targets MEK1/2 and KSR1. This will be broadly applicable for target identification strategies and to identify protein complexes of future molecular glues. This study is well-done and rigorously performed and should be accepted as is.

We were thrilled to hear that Reviewer 1 was enthusiastic about our study, and hope that they remain enthusiastic about our revision, which we believe has expanded the scope and utility of the work.

Reviewer #2 (Remarks to the Author):

In this paper, the authors developed a tool to identify proteins surrounding a small molecule. Their tool requires two components, a chemical containing the small molecule conjugated to the FKBP12(F36V) binding compound ortho-AP1867 and miniTurbo fused to FKBP12(F36V). Introduction of the chemical leads to binding of both the small molecule's target and miniTurbo, enabling proximity labeling of its vicinity. The authors demonstrated their tool to study the interactors of (+)-JQ1 and trametinib. They performed western blotting for both to demonstrate enrichment of known interactors and proteomics for (+)-JQ1. They demonstrated off-competition of the interactors with addition of the unconjugated small molecule.

We thank Review #2 for their astute and well-thought-out considerations of our manuscript. We believe the changes we have made to the document to address the concerns of Review #2 have markedly improved the overall work. Below we address each individual question and highlight where we have amended the manuscript to address these concerns when able. In circumstances where we were not able to directly address questions or concerns with data we have provided a considered response updated the manuscript text where appropriate.

I recommend major revisions for the following reasons:

- The authors should address if their system alters the behavior of the small molecule itself and answer:
1. How does conjugation to ortho-AP1867 affect permeability compared to the small molecule itself?

This is an excellent and important question, as solubility and cell entry of the bifunctional ligand will be imperative to the applicability of this method. We had attempted to address this question previously in Supplemental Figures 1D-F and 7A using cellular target engagement assays. However, more could be done to validate and compare permeability of these compounds as compared to their untagged counterparts. To provide a definitive benchmark for the cell permeability of the small molecules, we turned to our FKBP12^{F36V} target engagement assay, and prepared reference compound JWJ-01-341 consisting of an N-Methyl derivative of FKBP12^{F36V} binder orthoAP1867. Using this as a control for a permeable small molecule binder of FKBP12^{F36V}, we show that the bifunctional compounds are often less cell permeable than JWJ-01-341, as indicated by their less potent engagement of FKBP12^{F36V} as compared to JWJ-0-341. These data are now also included in Figure S6 (Alisertib) and Figure S7 (Trametinib/Trametiglu).

Reviewer Figure 1 | Cell permeability of bifunctional molecules. A. Assay schematic of FKBP12^{F36V} cellular target engagement assay, adapted from Nabet *et al.* B. Chemical structure of JWJ-01-341-1. C. Cellular target engagement of example molecules relative to control JWJ-01-341. To evaluate the cellular target engagement of the named small-molecule target protein, we performed a CETSA assay, evaluating relative abundance of soluble proteins by immunoblot. We used published melting curves for BRD4 and Aurka, and experimentally determined the cellular melting temperature of MEK1/2 (Figure S7J).

The FKBP12^{F36V} target engagement assays cannot be performed using the unmodified small molecule ligands-of-interest. To evaluate these directly we instead measured target engagement with a cellular thermal shift (CETSA) assay to directly compare the bifunctional and parental ligands against their named target protein. We identified the lowest temperature for each target protein where > 90% loss of signal is observed (47.5 °C for BRD4, 51.5 °C for Aurka and 62°C for MEK1/2), and assayed the bifunctional molecules ability to stabilize the target protein relative to the parental, untagged ligands. Here, we find that the cellular target engagement of the major targets is maintained, but in the cases of poorly permeable molecules, such as Cpd 5, are reduced relative to parental inhibitor controls. These effects may be due to both cell permeability differences and differences in affinity, but still demonstrate the capacity of these ligands to enter the cell and stabilize the known molecular target of the conjugated small molecule.

Reviewer Figure 2 | Thermal melting curve of MEK1/2 in HEK293T cells. A. Immunoblot analysis of soluble proteins isolated from HEK293 cells incubated at the indicated temperature. B. Normalized abundance quantified from A. (**Supporting Figure 7J**)

Reviewer Figure 3 | CETSA assay of bifunctional molecules. A. Thermal Stabilization of MEK1/2 in HEK293 cells by inhibitors and bifunctionals. B. Average thermal stabilization of MEK1/2 across two biological replicates. C. Thermal Stabilization of BRD4 in HEK293 cells by inhibitors and bifunctionals. D. Thermal Stabilization of Aurka in cells by inhibitors and bifunctionals. A.-B. These data are now presented in the manuscript in **Supporting Figure 7K**.

Thus, tagging with ortho-AP1867 does reduce compound permeability for a subset of our engineered molecules, as may be expected by the significant chemical addition. However, all bifunctional ligands produced maintain solubility sufficient for direct binding, stabilization, and also labeling of known targets and interacting partners of these ligands.

Could it affect permeability not just through the plasma membrane but also other organelle membranes?

We thank the reviewer for this question, as it highlighted a gap we had not fully considered in our methods ability to perform "unbiased" labeling of cellular drug targets. The permeability of small molecules into membrane enclosed organelles, such as the endoplasmic reticulum, Golgi apparatus, nucleus, mitochondria, lysosomes, endosomes, and peroxisomes, is rarely reported for chemical probes and drug candidates, and we struggled to identify a quantitative assay that would allow us to compare the small molecules organelle permeability and accumulation across all organelles at the throughput required to test all the bifunctional and parental the molecules reported in this manuscript. We note that in analysis of our bulk mass spectroscopy data for DMSO conditions where BioTAC constructs and biotin are present but no guiding bifunctional molecule, biotinylated proteins known to localize within every organelle were identified. However, it is unclear whether these are being passively labeled by our enzyme within these compartments, or if labeling of these components is occurring during co-translation with the miniTurboFKBP12^{F36V} enzyme. Additionally, this observation does not resolve the question of ligand permeability and targeted labeling to these compartments. We have highlighted this uncertainty within the manuscript and further noted compartments for which we have confirmed target engagement.

We do note, and have expanded on, our observation that we are able to label proteins within the nucleus, a cellular compartment rich in drug targets, with a bifunctional BioTAC ligand. Here, we demonstrate that Cpd 1 can colocalize miniTurboFKBP12^{F36V} with BRD4 in the nucleus (Supporting Figure 2) in a manner dependent on BRD4 engagement. These data indicate that Cpd 1 is able to access the nuclear compartment either in complex with or independently of the miniTurboFKBP12^{F36V} enzyme.

Supporting Figure 2 | Colocalization of BioTAC constructs and BRD4. Representative maximum intensity projections of confocal microscopy images of HEK293 cells transiently transfected with miniTurboFKBP12^{F36V} and treated with the indicated compound for 30 min. Scale bar = 10 μm. Blue: nucleus (DAPI), green: HA (miniTurboFKBP12^{F36V}), red: BRD4, white: colocalization, ND: no colocalization detected. 10 Z-stacks were collected per image and processed using colocalization thresholding between BRD4 and HA.

We also wish to highlight that our (+)-JQ1 bifunctionals engaged nuclear-localized BRD4 in live cells by CETSA (see **Reviewer Figure 3**) comparably to (+)-JQ1, indicating that nuclear permeability is not compromised. This is further supported by the fact that the majority of hits identified by proteomics enriched by Cpd 1 and out competed by (+)-JQ1 are nuclear localized proteins (**Figure 2, Figure 3**), with the exception of GSTK1, a glutathione transferase responsible for small molecule metabolism located in peroxisomes (<https://www.proteinatlas.org/ENSG00000197448-GSTK1/subcellular>). Our general conclusion remains that we can access the cytoplasm and nucleus with our engineered chemical ligands, compartments containing a large proportion of drug targets. We remain unable to clearly define if other subcellular compartments can be accessed by this toolset, as this would require bonafide tool compounds validated to interact with proteins exclusively located in these compartments to adequately answer that question. We have amended our description of the labeling scope of our system to reflect this uncertainty.

2. Is the difference in permeability between cpd1 and cpd2 due to the difference in permeability of (+)-JQ1 and trametinib?

The cell permeability of the orthoAP1867 containing bifunctional molecules depends on their overall physicochemical properties, which in turn, depend on the identity of the conjugated ligand and the linker. Hence, variability in cell permeability is expected. However, we demonstrate that even bifunctionals with reduced cell permeability, such as Cpd 4 and Cpd 5, achieve successful targetID and interactomeID by unbiased BioTAC proteomics.

3. Does the addition of the ortho-AP1867 handle and recruitment of miniTurbo affect the small molecule's behavior (ie increase steric hindrance that could block interactors)?

We thank the reviewer again for this extremely valuable question. This was not something we had directly measured previously and is a notable caveat we must highlight for this method. Certainly, the proximity of the miniTurboFKBP12^{F36V} enzyme and chemical linker could alter the accessibility of the target protein and thus the binding complex members associated with it. To address this a linker attached to an alternate attachment site, that alters the approach vector of miniTurboFKBP12^{F36V}, would be necessary. To investigate how the linker attachment site and miniTurbo-FKBP12^{F36V} recruitment vector alters the detected interactome of (+)-JQ1, we synthesized a series of bifunctional (+)-JQ1 molecules conjugated at the 4-phenyl position, and performed immunoblot confirmation of BRD4 enrichment with dose-dependent off-competition with (+)-JQ1 (Figure 3B, S5A-B).

Using Cpd 2 we performed BioTAC global proteomics analysis at the 4 h time point, and compared the hits with those obtained in matched experiments with Cpd 1. Here, we saw consistent enrichment of BRD2, BRD3 and BRD4 across datasets. Further, we found a core set of known BRD4 interactors were identified in both datasets, including ATAD5, KMT5B, NSD3 and UBTF, alongside detoxification enzyme GSTK1. We also observed a significant number of non-overlapping hits in each dataset. These non-overlapping hits could be a result of the altered ligand or steric clash with the BioTAC enzyme, or of an altered interactome of 4-phenyl functionalized (+)-JQ1, but were all known BRD2/3/4 interactors or chromatin-associated proteins. The identification of a divergent interactome is consistent with recent reports, which showed 4-phenyl functionalization of (+)-JQ1 (as in Cpd 2) enables recruitment of proteins such as DCAF16, which were not recruited by tert-butyl ester derivatives (as in Cpd 1).¹⁻³ The interaction interface for many of these associated complex members is distal to the ligand binding site, suggesting that the difference in complex association may be due to ligand chemotype, rather than steric inhibition of our complex, and that core components were still identifiable between both labeling systems.

Hence, we conclude surveying multiple linker attachment sites can expand the detected interactome by the BioTAC System, likely via a combination of altering proximity-labeling approach vector and altering ligand structure, and we have now highlighted this in the manuscript.

These data are now included in the manuscript in Figure 3 and Supporting Figure 5.

Figure 3 | Linker Exit Vector Diversification Identifies Core and Extended Interactors. A. Chemical structure of (+)-JQ1 bifunctional with alternative linker attachment site. **B.** Immunoblot analysis of BRD4 enrichment following treatment of HEK293 cells transiently transfected with miniTurboFKBP12^{F36V} with the indicated compounds and 100 μM biotin at the 30 min timepoint. Data representative of $n = 2$ biologically independent experiments (SI Figure 5 B). **C.** Venn diagram showing statistically significant hits at the 30 min time point from DIA BioTAC experiments using Cpd 1 (blue) or Cpd 2 (purple). Overlapping hits are listed below. **D., E.** Proteomics using data-independent acquisition methods of streptavidin-enriched biotinylated proteins isolated from HEK293 cells transiently transfected with miniTurboFKBP12^{F36V} and treated with DMSO, 1 μM Cpd 2, plus the indicated compounds and 100 μM biotin for 30 min, demonstrating enrichment and competition of known direct targets and complexed proteins. Only proteins with P -value < 0.05 in both conditions depicted. Complete datasets in Table S1. Known targets shown dark blue, known interactors or proteins participating in the same biological process shown light blue.

Supporting Figure 5 | Optimization of alternate-linker (+)-JQ1 orthoAP1867 bifunctional molecules. A. Chemical structure of JWJ-01-359. B. Immunoblot analysis of BRD4 enrichment following treatment of HEK293 cells transiently transfected with miniTurboFKBP12^{F36V} and treated with the indicated compound and 100 μM biotin at the 30 min timepoint.

4. How do the dose curves of the conjugated small molecule and small molecule itself compare?

Thank you for this question. This is an important point which we failed to clearly highlight in our original manuscript. For (+)-JQ1, extensive data demonstrating that conjugation of a linker at the attachment sites used in this study have minimal effects on primary target binding affinity. Similarly, we⁴ and others⁵ have shown that conjugation of a linker to Alisertib at the sites used in this study results in a modest decrease in Aurka binding. For Trametinib, the linker attachment site had been reported by Khan *et al.*⁶ These data motivated our selection of attachment sites. However, for Trametigluce, no linker-attached derivatives have been reported, and additionally, the cell target engagement may be affected by the cell permeability. Therefore, we performed CETSA analysis of our bifunctionals and compared the thermal stabilization to parental inhibitors (see **Reviewer Figures 2 and 3**). For BRD4 we confirmed comparable stabilization by Cpd 1 and (+)-JQ1 at 1 μM and 10 μM and modestly reduced stabilization by Cpd 3 compared to Alisertib in line with published findings and cell permeability differences. Happily, comparable stabilization of MEK1/2 was observed by Trametinib, Cpd 4, Trametigluce and Cpd 5, though absolute stabilization varied between replicates, compound ranking remained the same. These data are shown in reply to point 1. CETSA analysis for MEK1/2 is now included in **Figure S7 J-K**. We have noted in the manuscript discussion that careful characterization of conjugated small molecules should be performed to evaluate changes in observed permeability by FKBP12^{F36V} target engagement, or if known in target binding or phenotype, to better optimize their use as labeling tool compounds.

• The authors should also clarify the cellular context of the tool and address the questions below:
5. In the cell, is the chemical in excess or is miniTurbo-FKBP12(F36V) in excess?

Stoichiometry is an important consideration for this system. Based on our cellular target engagement assays (Reviewer Figure 1 and elsewhere in the manuscript) we believe the enzymatic construct is in excess of ligand. To investigate this further, we performed a titration of Cpd 4 across a 10-fold range

of concentrations (1 to 10 μM) and evaluated MEK1 enrichment by immunoblot. This experiment further supports that the enzymatic construct is in excess, with a range of small molecule concentrations resulting in equivalent MEK1/2 labeling at each time point. We have clarified the importance of this stoichiometry in the manuscript and added these data in support of that conclusion.

Supporting Figure 7 B | Immunoblot analysis of MEK1/2 enrichment following treatment of HEK293 cells transiently transfected with miniTurboFKBP12^{F36V} with JWJ-01-280-1 and 100 μM biotin at the indicated timepoint.

6. If miniTurbo-FKBP12(F36V) is in excess, is there noise from free, unassociated miniTurbo-FKBP12(F36V) that could biotinylate its surrounding?

Yes! In the presence of biotin, the constitutively active miniTurboFKBP12^{F36V} constructs biotinylate the cellular proteome, and this can be seen in the DMSO control lanes of all blots, where low level biotinylation and therefore enrichment is observed for most protein targets. It can also be observed in all proteomic datasets, where thousands of proteins are pulled down and identified in the DMSO-treated samples. The small molecules do not affect miniTurboFKBP12^{F36V} activity, but rather, affect the proteins proximal to the miniTurboFKBP12^{F36V}. In the presence of a bifunctional such as Cpd 1, the small molecules binders (e.g. BRD4) will be recruited to the miniTurboFKBP12^{F36V} construct and therefore biotinylated at a higher rate than background labeling, in a manner that is dependent on ligand binding and that can be outcompeted by saturating concentration of the parental ligand. These control conditions (Cpd 1 vs DMSO, Cpd 1 vs 10x (+)-JQ1, Cpd 1 vs 20x (+)-JQ1), combined with our hit-calling criteria which mandates statistically significant enrichment vs DMSO, and statistically significant off-competition by free ligand, reveal which biotinylation events are ligand dependent and which are background labeling. To illustrate how these data look plotted on traditional volcano plots where only 2 conditions are compared, we include example volcano plots for one of the presented datasets in Figure S4A-C, which correspond to the multiple comparison plots depicted in Figure 2A. All raw data is available on PRIDE, and complete processed datasets for all experiments can be found in Table S1.

Supporting Figure 4A-C | Volcano plots of proteomic experiments. Volcano plots showing all datapoints from mass-spectrometry based proteomic analysis of proteins enriched by streptavidin pulldown from HEK293 cells transiently transfected with miniTurbo-FKBP12^{F36V} and treated with 100 μ M Biotin, DMSO or 1 μ M of Cpd 1 \pm 10 μ M (+)-JQ1 for A. 30 mins. B. 60 mins. C. 4 hrs. Corresponding to Figure 2A. Points corresponding to BRD3 and BRD4 are highlighted blue.

We also now include in the methods section example streptavidin blots before and after enrichment, demonstrating robust biotinylation in all lanes, as expected (see below).

Does the expression of miniTurbo-FKBP12(F36V) have to be titrated to prevent excess, especially in cases where permeability of the chemical is limited? Could the authors provide streptavidin blots prior to enrichment?

An excellent question. No titration is needed with transient transfection, as excess miniTurbo-FKBP12^{F36V} is preferred to avoid the hook effect. However, high constitutive expression through stable cell line generation (a technique we attempted) does result in background incompatible with this method. We now highlight this caveat in the manuscript discussion. As with all miniTurbo proximity labeling experiments, background labeling is significant and must be accounted for using experimental controls. Even in the absence of a labeling enzyme a number of proteins enrich variably as naturally

biotinylated proteins. As a demonstration, we now provide streptavidin blots for an example experiment before enrichment, flow through, and post enrichment in **Supporting Figure 9C**, in the methods section, to aid researchers applying our method in diverse biological contexts.

Supporting Figure 9 | Additional optimization data. A. Immunoblot analysis of MEK1/2 and KSR1 enrichment following treatment of indicated cells transiently transfected with miniTurboFKBP12F36V and treated with JWJ-01-280 and 100 μ M biotin at the 4 hr timepoint. B. Immunoblot analysis of BioTAC construct transfection efficiency in A549 cells transiently transfected with miniTurboFKBP12F36V under the indicated conditions. C. Immunoblot analysis of global biotinylation in input (I), flowthrough (F), and enrichment (E) samples following treatment of HEK293 cells transiently transfected with miniTurboFKBP12F36V and treated with JWJ-01-280 and 100 μ M biotin at the 4 hr timepoint.

Our titration experiments further highlight that labeling time is more important than compound dose for a given cell line, i.e. extended labeling time can be used to increase on-target labeling signal. These data are now presented in **Supporting Figure 7B**.

Supporting Figure 7 B | Immunoblot analysis of MEK1/2 enrichment following treatment of HEK293 cells transiently transfected with miniTurboFKBP12^{F36V} with JWJ-01-280-1 and 100 μ M biotin at the indicated timepoint.

To evaluate the effect of biological variables, including construct expression level, we examined the BioTAC system in 3 cell lines, and found that robust results were obtained in HEK293 and HCT116 cells (**Supporting Figure 9A, C**). In cell lines where expression of the miniTurboFKBP12^{F36V} construct is very low due to inefficient transfection (e.g. A549), and compound cell permeability is very low (e.g. Cpd 4, and PEG3 derivative JWJ-01-295), then markedly reduced enrichment of MEK1/2 is observed.

In this case, optimization of the transfection to ensure sufficient expression would be expected to increase the signal. However, the requirement for efficient transfection is true of all plasmid-based systems, and can be achieved in a straightforward manner (e.g. Supporting Figure 9C). We do not believe that a titration would be required for each compound, but rather that a minimal expression level would be needed to see robust signal at short time points.

7. Where does miniTurbo-FKBP12(F36V) localize in the cell? If it is in the cytosol, and the drug goes to the nucleus, will miniTurbo-FKBP12(F36V) binding impede the drug's ability to reach its target? Or perhaps miniTurbo-FKBP12(F36V) won't be efficiently loaded on to the chemical in the nucleus?

8. Imaging miniTurbo-FKBP12(F36V) by introducing an epitope tag would be nice and could potentially show differences in localization with and without chemical treatment and with different chemical treatments. Since cpd1 detects BRD4 which is annotated as nuclear and cpd2 detects KSR1 which is not annotated as nuclear, perhaps localization of miniTurbo-FKBP12(F36V) changes depending on whether it is treated with cpd1 or cpd2.

Thank you for these excellent suggestions. We had in fact installed an HA tag to our BioTAC enzymatic construct to measure expression level and can also use this to address these important questions. We performed confocal microscopy on fixed, stained cells expressing HA-miniTurboFKBP12^{F36V} treated with either DMSO, Cpd1, or Cpd1 + 10x (+)-JQ1, and quantified co-localization of HA-tagged miniTurboFKBP12^{F36V} and BRD4. In untreated cells HA-miniTurboFKBP12^{F36V} is predominantly in the cytosol, but after Cpd 1 addition HA-miniTurboFKBP12^{F36V} translocated to the nucleus, and co-localized with BRD4. This co-localization was completely abrogated by pretreatment with excess (+)-JQ1. This is consistent with a recent preprint from the Schreiber lab that also demonstrated similar molecules can cause nuclear localization of GFP-FKBP12^{F36V} constructs.⁷ In addition, this agrees with our observation that hits identified by (+)-JQ1 BioTACs Cpd 1 and Cpd 2 are predominantly nuclear localized proteins.

Supporting Figure 2 | Colocalization of BioTAC constructs and BRD4. Representative maximum intensity projections of confocal microscopy images of HEK293 cells transiently transfected with

miniTurboFKBP12^{F36V} and treated with the indicated compound for 30 min. Scale bar = 10 μ m. Blue: nucleus (DAPI), green: HA (miniTurboFKBP12^{F36V}), red: BRD4, white: colocalization, ND: no colocalization detected. 10 Z-stacks were collected per image and processed using colocalization thresholding between BRD4 and HA.

Figure 2B | The BioTAC system enables rapid, accurate small molecule interactome-ID. B. Proteomics using data-independent acquisition methods of streptavidin-enriched biotinylated proteins isolated from HEK293 cells transiently transfected with miniTurboFKBP12^{F36V} and treated with the indicated compounds and 100 μ M biotin for 30 min, demonstrating enrichment and competition of known direct targets and complexed proteins. High-confidence hits are defined as those that are enriched > 2-fold in both Cpd 1/ DMSO and Cpd 1 / Cpd 1 + 10x (+)-JQ1, where $P < 0.05$, plotted upper-right quadrant. Only proteins with P -value < 0.05 in both conditions depicted. Complete datasets in Table S1. Known targets shown dark blue (labelled in left plot), known interactors or proteins participating in the same biological process shown light blue (labelled in right plot).

We next compared these effects with those induced by Cpd 4, imaging HA tagged miniTurboFKBP12^{F36V} and MEK1/2. Here, we observed cytosolic localization of miniTurboFKBP12^{F36V} in the presence and absence of Cpd 4 and Cpd 5, consistent with the cytosolic localization of MEK1/2. No nuclear construct expression was observed, as measured by correlation analysis vs nuclear stain DAPI, and in contrast to observations with Cpd 1 for BRD4.

Supporting Figure 8 | Colocalization of BioTAC constructs and MEK1/2. Confocal microscopy of HEK293 cells transiently transfected with miniTurboFKBP12^{F36V} and treated with the indicated compound for 30 min. Blue: nucleus (DAPI), green: HA (miniTurboFKBP12^{F36V}), red: MEK1/2, ND no correlation detected between HA and DAPI indicating a lack of miniTurboFKBP12^{F36V} in the nucleus. Scale bar = 10 μm.

Together, these studies demonstrate that the BioTAC system can be used to identify small molecule interacting targets in the nucleus and cytosol and that while the unbound miniTurboFKBP12^{F36V} construct is found predominantly in the cytoplasm it can be rapidly localized to the nucleus with the correct bifunctional ligand.

9. If the drug's target is in the mitochondria or secretory system, I suspect the tool would not work and you would need prior knowledge about the drug to target miniTurbo-FKBP12(F36V) to those organelles first.

We agree that the BioTAC system is currently unsuitable for evaluating extracellular, secreted, or luminal mitochondrial proteins. We have updated the text to reference intracellular target-ID and intercellular interactome-ID.

We struggled to evaluate the ability of the BioTAC system to label proteins inside the mitochondria, due to the limited number of options for targeting large proteins for mitochondrial import (as opposed to tethering at cytoplasmic face of the mitochondrial membrane). Using recently a published tag capable of delivering GFP to the mitochondrial lumen we attempted to deliver a truncated BRD4 peptide constituting one of the bromodomains (BD2), which could be targeted by Cpd 1, to the mitochondria using a pCNDA3.1 mitotag-EGFP-BD2 construct. . We failed to observe robust BioTAC labeling of this construct with Cpd 1, however, interpretation of these preliminary results is limited by the lack of enrichment seen in matched GFP-BRD4-BD2 cytosolic controls, indicating our approach was not suitable for drawing definitive conclusions about mitochondrial target labeling (*data not shown*). AlphaFold simulations of this construct curiously demonstrated a stable interaction between the BD2 domain and the GFP beta-barrel at the interface of the BD2 ligand binding site, which may have occluded our ability to label this construct under any condition. Unfortunately, given the size limits

of proteins that can be imported into the mitochondrial lumen by the pCND3.1 mitotag system, we were unable to attempt this experiment with larger BRD4 constructs.

We conclude that the BioTAC System is only rigorously benchmarked for cytosolic and nuclear compartments, and have updated the text to indicate this. Since cytosolic and nuclear localized proteins represent approximately 10,670 of the 13,145 proteins of the human genome⁸ the BioTAC System will still find broad utility in uncovering the targets and interactomes of small molecules.

• On the (+)-JQ1 section,

10. Does the drug bind to BRD2/3/4 one at a time or does BRD2/3/4 complex with each other?

We apologize for our lack of clarity on this topic. (+)-JQ1 binds BRD2, BRD3, and BRD4 independently and each protein contains two ligand binding sites, bromodomain 1 and bromodomain 2. These interactions have been extensively studied *in vitro* and in cells. (+)-JQ1 is capable of binding each protein in the absence of the others.⁹ We have more clearly articulated this in the text for clarity.

11. If the drug binds to one BRD at a time, could the difference in enrichment be due to difference in affinities or difference in expression of BRD2/3/4?

This is an excellent question and one we did not fully address upon seeing variable detection of BRD2, BRD3, and BRD4 in the original manuscript. (+)-JQ1 binds potently, and fairly equivalently, to the bromodomains of BRD2, BRD3 and BRD4 (K_d 128 nM – 49 nM).⁹ Our inability to detect BRD2 in the 30 min time point caused us to also question why BRD2 was detected only at longer incubation time points. We hypothesized that BRD2 was being labeled, but that unique tryptic peptides were not being detected by mass spectrometry. BRD2 shares >50% of its total sequence identity with BRD3 and BRD4, and nearly half of its tryptic peptides that could be hypothetically detected on a mass spectrometer are observed identically or with high similarity to BRD3 and BRD4 peptides, meaning high sensitivity is required to observe the unique BRD2 peptides with high confidence, particularly if only a subset of BRD2 is labeled. We investigated if ToF mass spectrometry, which has increased sensitivity,¹⁰ and the use of data-independent acquisition which is reported to provide comprehensive proteome coverage and quantitation¹⁰, would enable robust detection of BRD2 as an interactor of (+)-JQ1. In these experiments we robustly detect BRD2, BRD3 and BRD4 enrichment in the presence of Cpd 1, and off-competition by (+)-JQ1, at the shortest 30 min timepoint. We have now added these experiments to the manuscript and included discussion in the main text.

Figure 2B | The BioTAC system enables rapid, accurate small molecule interactome-ID. B. Proteomics using data-independent acquisition methods of streptavidin-enriched biotinylated proteins isolated from HEK293 cells transiently transfected with miniTurboFKBP12^{F36V} and treated with the indicated compounds and 100 μM biotin for 30 min, demonstrating enrichment and competition of known direct targets and complexed proteins. High-confidence hits are defined as those that are

enriched > 2-fold in both Cpd 1/ DMSO and Cpd 1 / Cpd 1 + 10x (+)-JQ1, where $P < 0.05$, plotted upper-right quadrant. Only proteins with P -value < 0.05 in both conditions depicted. Complete datasets in Table S1. Known targets shown dark blue, known interactors or proteins participating in the same biological process shown light blue.

Thus, equipment sensitivity, not labeling efficiency, is the primary limiting factor dictating the necessary duration of labeling to ligand binding partners and interactors. However, these targets can be enriched with further labeling. Overall, the targets and interactors identified by the BioTAC System coupled to mass spectrometry were reliable and performed well against benchmarking datasets regardless of the instrumentation and data acquisition method used, which we believe adds to its utility as a widely accessible method.

• On the trametinib section,

12. The authors do not show anything to support its properties as a molecular glue. Their data does not show the differentiation in off-rate when MEK1 and KSR1 are both engaged versus MEK1 alone. The difference in the off-compete labeling of the MEK1 and KSR1 just demonstrates they bind with different affinities but not whether their affinities change depending on the presence of each other. Perhaps they should do knockout or knockdown of one of the components to show the difference in off-compete labeling when one of the interactors is absent.

Thank you for highlighting this discrepancy between our statements in the document and the resultant data. We failed to clearly describe the rationale behind these experiments, the background on this compound, and our intentions with our initial experiments in sufficient detail. Trametinib is a well-defined molecular glue that has been previously characterized for its capacity to stabilize the interaction between MEK1/2 (binding target) and KSR1 (recruited to trametinib bound MEK1/2). We have added text to clarify this background. We had intended to ask if the BioTAC System could be used to observe a known glue-enhanced interacting molecule, the detection of which is frequently impaired by low signal to noise in other unbiased ligand interaction mapping technologies.

Although not our original intention, the reviewer's questions spurred us to ask if the BioTAC System can be used to demonstrate molecular glue pharmacology. First, we considered the knockout experiments suggested by the reviewer. Unfortunately, dual knockout of MEK1 and MEK2 is not tolerated by cells, with conditional double knockouts of the paralogs MEK1/2 reported to result in cell death due to their essential roles in cells.¹¹ Therefore, we could not attempt the reviewer requested experiment to investigate KSR1 labeling and off-compete in the absence of MEK1/2, though this would be precisely the way to address this question under more auspicious conditions. We therefore sought to identify alternative methods to use BioTAC as a tool to investigate molecular glue pharmacology of essential protein targets.

To investigate the ability of the BioTAC system to differentiate glue pharmacology from inhibitor pharmacology we developed mKSR1 constructs harboring a W781D mutation. W781 lies at the molecular glue interface of KSR1 and MEK1/2 and Khan *et al* have shown that the KSR1^{W781D} mutation abrogates Trametinib-induced complexation. We then used the BioTAC system to evaluate if enrichment of KSR1 was reduced in cell lines overexpressing mKSR1 W781D as compared to wild-type. We used Cpd 4 (Trametinib-derived bifunctional, **Reviewer Figure 4**) and developed 2 new compounds; Cpd 5 (Trametigluce-derived bifunctional, **Reviewer Figure 5**) to model a second MEKi molecular glue with a related interactome profile to trametinib, and Cobimetinib-derived bifunctional JWJ-01-355, which does not have reported glue pharmacology and instead is an allosteric MEK1/2 inhibitor (**Reviewer Figure 6**). We used an anti-FLAG antibody to detect KSR1 pulldown and hence in these blots no signal is seen in the absence of construct transfection.

Consistent with expected results, we observed enrichment and off-competition of wtKSR1, and reduced enrichment by KSR1^{W781D} with Cpd 4 and Cpd 5, and no enrichment of KSR1 by JWJ-01-355

relative to DMSO. However, we also observed a high background labeling of KSR1 in all the DMSO conditions in this experiment, that may be due to either experimental errors or an artefact of the variable overexpression levels observed in this experiment, and the high KSR1 to MEK1 ratio. Furthermore, JWJ-01-355 was less effective at enriching MEK1/2 than Cpd 4 and Cpd 5.

Reviewer Figure 4 | KSR1 mutation at the MEK1 KSR1 interface abrogates enrichment of KSR1 by Cpd 4. Immunoblot analysis of mKSR1 following treatment of HEK293 cells transiently transfected with miniTurbo-FKBP12^{F36V} and mKSR1, with the indicated compounds and 100 μM biotin at the 4 h timepoint, and streptavidin-based enrichment, showing successful enrichment and competition with trametinib. KSR1 quantification (anti-FLAG) normalized to input KSR1 (anti-FLAG) overexpression levels.

Reviewer Figure 5 | KSR1 mutation at the MEK1 KSR1 interface abrogates enrichment of KSR1 by Cpd 4. Immunoblot analysis of mKSR1 following treatment of HEK293 cells transiently transfected with miniTurbo-FKBP12^{F36V} and mKSR1, with the indicated compounds and 100 μM biotin at the 4 h timepoint, and streptavidin-based enrichment, showing successful enrichment and competition with trametinib. KSR1 (anti-FLAG) quantification normalized to input KSR1 (anti-FLAG) overexpression levels.

Reviewer Figure 6 | JWJ-01-355 is unable to enrich KSR1. A. Chemical structure of cobimetinib derived bifunctional. B. Cellular FKBP12^{F36V} target engagement of JWJ-01-355. C. Immunoblot analysis of mKSR1 following treatment of HEK293 cells transiently transfected with miniTurbo-FKBP12^{F36V} and mKSR1, with the indicated compounds and 100 μM biotin at the 4 h timepoint, and streptavidin-based enrichment, showing unsuccessful enrichment and competition with Cobimetinib. KSR1 (anti-FLAG) quantification normalized to input KSR1 (anti-FLAG) overexpression levels.

Thus, exogenous overexpression, even of impaired mutant variants, is unlikely to be a route toward discriminating molecular glue binding events which in this case serve to enhance interactions between molecules rather than create de novo binding interfaces. Unfortunately, time constraints for returning a revised manuscript (3 months) prevented us from optimizing these experiments further to generate more conclusive data. Therefore, we have instead updated the text to clarify that the BioTAC system is a useful tool for generating hypotheses about compound mechanism of action. As with all new experimental findings, orthogonal validation is recommended when applied to uncharacterized compounds.

- This tool could be more robustly validated and explored.

13. Western blotting of MS hits besides BRD4

We agree that further demonstrations of the BioTAC system as a way to characterize ligand engagement with target complexes would strengthen the paper. We have now added additional examples to validate the tool more robustly, as suggested by the reviewer. These include MS and western blotting of a novel derivative of Alisertib, a known binder of Aurka (Figure 4, Figure S6).

Figure 4 | The BioTAC System is an Extensible Strategy for Live-Cell Target-ID. A. Chemical Structure of Cpd 3. B. Immunoblot analysis of AurkA enrichment following treatment of HEK293 cells transiently transfected with miniTurboFKBP12^{F36V} with the indicated compounds and 100 μM biotin at the 4 h timepoint. C., D. Proteomics using data-independent acquisition methods of streptavidin-enriched biotinylated proteins isolated from HEK293 cells transiently transfected with miniTurboFKBP12^{F36V} and treated with DMSO, 1 μM Cpd 3, plus the indicated compounds and 100 μM biotin for 4 h, demonstrating enrichment and competition of known direct targets and complexed proteins. Only proteins with P -value < 0.05 in both conditions depicted. Complete datasets in Table S1.

Supporting Figure 6 | Optimization of Alisertib-orthoAP1867 bifunctionals. A. Chemical structure of control compound JWJ-01-341, and alisertib bifunctionals. B. Immunoblot analysis of BRD4 enrichment following treatment of HEK293 cells transiently transfected with miniTurboFKBP12^{F36V} and treated with the indicated compound and 100 μ M biotin at the indicated timepoint. C. FKBP12^{F36V} cellular target engagement assays for Alisertib bifunctionals, data plotted as mean \pm S.D. of $n = 3$ technical replicates.

In addition, we synthesized Trametigluce-derived bifunctionals and performed MS, and follow up western blotting of MEK1/2 and KSR1 hits (Figure 5F-G, Figure S7F-K).

Figure 5F-G | The BioTAC system enables detection of non-degrader molecular glue interactions. F. Chemical structure of Trametinigluie bifunctional molecule JWJ-01-354-1 / Cpd 5. G. Proteomics using data-independent acquisition methods of streptavidin-enriched biotinylated proteins isolated from HEK293 cells transiently transfected with miniTurboFKBP12^{F36V} and treated with DMSO, 1 μ M Cpd 5, plus the indicated compounds and 100 μ M biotin for 4 h, demonstrating enrichment and competition of known direct targets and complexed proteins. Only proteins with *P*-value < 0.05 in both conditions depicted. Complete datasets in Table S1. Known targets shown dark blue, known interactors or proteins participating in the same biological process shown light blue.

Supporting Figure 7F-K | Optimization of MEK1 labeling by trametinib and trametinigluie bifunctional molecules. F. Chemical structure of JWJ-01-348. G., H. Immunoblot analysis of MEK1/2 and KSR1 enrichment following treatment of HEK293 cells transiently transfected with miniTurboFKBP12^{F36V} with the indicated concentration of the indicated compound and 100 μ M biotin for 4 h. I. FKBP12^{F36V} cellular target engagement assays for trametinigluie bifunctionals, data plotted as mean \pm S.D. of *n* = 3 technical replicates. J. CETSA analysis of MEK1/2 melting point. Data based on western blot quantification, normalized to the lowest temperature of 42 °C. K. Thermal stabilization of MEK1/2 by inhibitors and bifunctional derivatives. Data shown as mean of *n* = 2 replicates. HEK293 cells were treated with the indicated compounds at the indicated concentrations for X mins, followed by isolation and incubation at 62 °C. Data based on western blot quantification, normalized to DMSO treated controls. S.D. Standard deviation.

We have added additional Cpd 1 replicate BioTAC proteomics using ToF mass spectrometry and DIA, where we again successfully identified BRD2, BRD3 and BRD4 as interactors of (+)-JQ1, alongside known BRD4-interactome proteins, further confirming BioTACs reproducibility across different experimental setups (Figure 2).

Next, we combined all hits from these datasets and compared them to the interactome data generated by established BioID and AP/MS of BRD2, BRD3, BRD4 and BRDT in the presence and absence of (+)-JQ1 in HEK293 cells reported by Lambert *et al.* and found 41.7 % overlap with our hits. To identify the likelihood of generating a list of proteins with 41.7 % overlap with the Lambert *et al.* reference dataset, we performed bootstrap analysis by comparing the overlap between 5,000 sets of proteins generated at random from the human transcriptome, and show that 41.7% overlap is over 88 standard deviations away from random chance.

Figure 2C | Enrichment of reported BET protein interactome in hit dataset vs that expected by random chance.

Supporting Figure 5 | Optimization of alternate-linker (+)-JQ1 orthoAP1867 bifunctional molecules. A. Chemical structure of JWJ-01-359. B. Immunoblot analysis of BRD4 enrichment following treatment of HEK293 cells transiently transfected with miniTurboFKBP12^{F36V} and treated with the indicated compound and 100 μM biotin at the 30 min timepoint.

14. Proteomics or blotting with Cpd2 samples to identify other proteins that complex around MEK1 and KSR1

We agree, further validation of this ligand was warranted. We performed BioTAC coupled to mass spectrometry with both Cpd 4 (prev. Cpd 2) and newly developed Cpd 5 derived from Trametigluce (Figure 5E,G). We successfully identified MEK1, MEK2, and KSR1. We show that the BioTAC system can identify putative off targets, such as MEK5, as well as additional MEK1 complex members, such as BRAF and ARAF.

Figure 5 | The BioTAC system enables detection of non-degrader molecular glue interactions.

A. Schematic depicting how the BioTAC system can detect molecular glue interactions. B. Chemical structure of Trametinib bifunctional molecule JWJ-01-280-1/Cpd 2. C. Immunoblot analysis of MEK1 and KSR1 following treatment of HEK293 cells transiently transfected with miniTurboFKBP12^{F36V} with the indicated compounds/growth factors and 100 μM biotin at the 4 h timepoint, and streptavidin-based enrichment, showing successful enrichment and competition with trametinib. D. Immunoblot analysis of mKSR1 following treatment of HEK293 cells transiently transfected with miniTurbo-FKBP12^{F36V} and mKSR1, with the indicated compounds/growth factors and 100 μM biotin at the 4 h timepoint, and streptavidin-based enrichment, showing successful enrichment and competition with trametinib. Data representative of $n = 3$ replicates (see also Fig. S7 D-E.) In C, D two KSR1 isoforms are observed, produced by alternative splicing.¹² KSR1-L (102 KDa) corresponds to the expected MW of Uniprot Q8IVT5-1 (canonical sequence), KSR1-S (87 KDa) corresponds to the expected MW of variant with

residues 1-137 missing, Uniprot Q8IVT5-3, and -4. Our data indicate both isoforms can complex with trametinib-bound MEK1. E. Proteomics using data-independent acquisition methods of streptavidin-enriched biotinylated proteins isolated from HEK293 cells transiently transfected with miniTurboFKBP12^{F36V} and treated with DMSO, 1 μ M Cpd 4, plus the indicated compounds and 100 μ M biotin for 4 h, demonstrating enrichment and competition of known direct targets and complexed proteins. Only proteins with P -value < 0.05 in both conditions depicted. Complete datasets in Table S1. Known targets shown dark blue, known interactors or proteins participating in the same biological process shown light blue. F. Chemical structure of Trametigluce bifunctional molecule JWJ-01-354-1 / Cpd 5. G. Proteomics using data-independent acquisition methods of streptavidin-enriched biotinylated proteins isolated from HEK293 cells transiently transfected with miniTurboFKBP12^{F36V} and treated with DMSO, 1 μ M Cpd 5, plus the indicated compounds and 100 μ M biotin for 4 h, demonstrating enrichment and competition of known direct targets and complexed proteins. Only proteins with P -value < 0.05 in both conditions depicted. Complete datasets in Table S1. Known targets shown dark blue, known interactors or proteins participating in the same biological process shown light blue.

15. Using BioTAC to deorphanize drug targets

We absolutely agree that this is the next step in the application of the BioTAC technology. However, efficient use of this platform for that purpose will require that we identify drugs with unknown mechanism-of-action, and establish the relevant patient-derived disease models systems likely to express their target proteins in our laboratories, establish the BioTAC method in these model systems and create a number of new bifunctional ligands. The scope and scale of such a project is underway, but far from complete, and would substantially detract from our current demonstration that this toolset is sufficient to label drug bound protein complexes. We have, however, shown the BioTAC system can be used to generate novel hypotheses about the molecular basis of small molecule pharmacology, even for well studied compound.

First, we show that Cpd 1 induces labeling of IPO9, a nuclear import protein, a first step in deorphanizing the target that contributes to the active transport component of Cpd 1, and potentially shared with the mechanism of action of the recently reported Nuclear Import and Control of Expression (NICE) class of bifunctional molecules.⁷ Extensive orthogonal validation beyond the scope of this manuscript would be required to validate this, but it is included as an example of how BioTAC system can be a valuable tool in identifying candidates. We have now updated the text to include discussion of both this finding, and added to the discussion text to clarify the role of BioTAC system as a hypothesis generating tool.

Figure 2B | The BioTAC system enables rapid, accurate small molecule interactome-ID. B. Proteomics using data-independent acquisition methods of streptavidin-enriched biotinylated proteins isolated from HEK293 cells transiently transfected with miniTurboFKBP12^{F36V} and treated with the

indicated compounds and 100 μ M biotin for 30 min, demonstrating enrichment and competition of known direct targets and complexed proteins. High-confidence hits are defined as those that are enriched > 2-fold in both Cpd 1 / DMSO and Cpd 1 / Cpd 1 + 10x (+)-JQ1, where $P < 0.05$, plotted upper-right quadrant. Only proteins with P -value < 0.05 in both conditions depicted. Complete datasets in Table S1. Known targets shown dark blue, known interactors or proteins participating in the same biological process shown light blue.

Next, we show that chemically diverse (+)-JQ1 derived bifunctionals, Cpd 1 and Cpd 2, promote labeling of GSTK1, a glutathione transferase involved in small molecules detoxification, and that this is off competed by excess free (+)-JQ1. This data suggests that GSTK1 may be involved in the cellular metabolism of (+)-JQ1 and (+)-JQ1 derived molecules (Figure 1-2).

• Additional minor points:

16. Increasing labelling time doesn't mean it increases labeling radius, but more so the coverage of labeling in the vicinity of the enzyme increases.

Thank you for the clarification, the text has been updated.

17. Double check the intro, uMap uses blue, not UV light.

Thank you for spotting this omission, the text has been updated.

Reviewer #3 (Remarks to the Author):

In this manuscript, Tao et al. developed the BioTAC system using bifunctional compounds and miniTurbo-fused FKBP12(F36V) to map small molecule interactomes. By using this BioTAC system, they detected compound-dependent biotinylation of direct and indirect interacting proteins of BET inhibitors. They also detected biotinylation of MEK1/2 and KSR1, the target proteins of the molecular glue trametinib.

However, it is surprising that the authors performed only focused screens by Western blotting to benchmark molecular glue-induced interactome changes. Can this BioTAC system detect compound-dependent significant changes in MEK1, MEK2, and KSR1 by label-free or TMT based quantitative mass spectrometry of biotinylated proteins? This unbiased, proteome-wide screening is essential to demonstrate that the BioTAC system is a comprehensive profiling method.

We appreciate the reviewer's input, and agree, we did not adequately characterize the capabilities of the platform in our initial manuscript. We have now performed BioTAC coupled to mass spectrometry for both trametinib and trametiglu derived bifunctional molecules, where we successfully detect MEK1, MEK2, KSR1 alongside additional interactors such as ARAF.

Figure 5 | The BioTAC system enables detection of non-degrader molecular glue interactions. A. Schematic depicting how the BioTAC system can detect molecular glue interactions. B. Chemical structure of Trametinib bifunctional molecule JWJ-01-280-1/Cpd 2. C. Immunoblot analysis of MEK1 and KSR1 following treatment of HEK293 cells transiently transfected with miniTurboFKBP12^{F36V} with the indicated compounds/growth factors and 100 μM biotin at the 4 h timepoint, and streptavidin-based enrichment, showing successful enrichment and competition with trametinib. D. Immunoblot analysis of mKSR1 following treatment of HEK293 cells transiently transfected with miniTurbo-FKBP12^{F36V} and mKSR1, with the indicated compounds/growth factors and 100 μM biotin at the 4 h timepoint, and streptavidin-based enrichment, showing successful enrichment and competition with trametinib. Data representative of *n* = 3 replicates (see also Fig. S7 D-E.) In C, D two KSR1 isoforms are observed, produced by alternative splicing.¹² KSR1-L (102 KDa) corresponds to the expected MW of Uniprot Q8IVT5-1 (canonical sequence), KSR1-S (87 KDa) corresponds to the expected MW of variant with residues 1-137 missing, Uniprot Q8IVT5-3, and -4. Our data indicate both isoforms can complex with trametinib-bound MEK1. E. Proteomics using data-independent acquisition methods of streptavidin-enriched biotinylated proteins isolated from HEK293 cells transiently transfected with miniTurboFKBP12^{F36V} and treated with DMSO, 1 μM Cpd 4, plus the indicated compounds and 100 μM biotin for 4 h, demonstrating enrichment and competition of known direct targets and complexed proteins. Only proteins with *P*-value < 0.05 in both conditions depicted. Complete datasets in Table S1. Known targets shown dark blue, known interactors or proteins participating in the same biological process shown light blue. F. Chemical structure of Trametiglu bifunctional molecule JWJ-01-354-1 / Cpd 5. G. Proteomics using data-independent acquisition methods of streptavidin-enriched biotinylated proteins isolated from HEK293 cells transiently transfected with miniTurboFKBP12^{F36V} and

treated with DMSO, 1 μ M Cpd 5, plus the indicated compounds and 100 μ M biotin for 4 h, demonstrating enrichment and competition of known direct targets and complexed proteins. Only proteins with P -value < 0.05 in both conditions depicted. Complete datasets in Table S1. Known targets shown dark blue, known interactors or proteins participating in the same biological process shown light blue.

Minor

comments:

1. TurboID and miniTurbo should be distinguished. Avoid using the terms miniTurboID or mTurbo.

Thank you for spotting this mistake, the text and figures have been updated.

2. The differences among Figure 1C, Figure S1I, and Figure S1J should be more clearly defined.

Thank you for pointing out the lack of clarity, the figure captions have been updated to clarify that the supporting figure panels are the biological replicates of the experiment in 1C, provided to show the reproducibility and expected variability in the experiments.

3. P-TEFb, TFIID, NuRD, and H4 should be plotted in Figure 2A.

We now plot representative known BRD4-interacting proteins in Figure 2B, 20x off-compete comparison. We also updated the text to include the protein names and not just the protein complexes for clarity.

Figure 2B | The BioTAC system enables rapid, accurate small molecule interactome-ID. B. Proteomics using data-independent acquisition methods of streptavidin-enriched biotinylated proteins isolated from HEK293 cells transiently transfected with miniTurboFKBP12^{F36V} and treated with the indicated compounds and 100 μ M biotin for 30 min, demonstrating enrichment and competition of known direct targets and complexed proteins. High-confidence hits are defined as those that are enriched > 2 -fold in both Cpd 1/ DMSO and Cpd 1 / Cpd 1 + 10x (+)-JQ1, where $P < 0.05$, plotted upper-right quadrant. Only proteins with P -value < 0.05 in both conditions depicted. Complete datasets in Table S1. Known targets shown dark blue, known interactors or proteins participating in the same biological process shown light blue.

4. Line 76, Figure S1d-e to Figure S1d-f.

Updated.

5. Page 1 of Supporting Information, add “Supporting Figure 6 | Uncropped blots7”

6. Figure S6B, upper, positions of markers may be wrong.

The uncropped western blot file has been double checked, updated, and marker lanes are now included and annotated to improve clarity. The uncropped blots are now in found the supporting data file to comply with Nature Communications formatting guidelines.

Response to Reviewer References

- 1 Oliver, H. *et al.* An intramolecular bivalent degrader glues an intrinsic BRD4-DCAF16 interaction. *bioRxiv*, 2023.2002.2014.528511, doi:10.1101/2023.02.14.528511 (2023).
- 2 Li, Y. D. *et al.* Template-assisted covalent modification of DCAF16 underlies activity of BRD4 molecular glue degraders. *bioRxiv*, doi:10.1101/2023.02.14.528208 (2023).
- 3 Shergalis, A. G. *et al.* CRISPR Screen Reveals BRD2/4 Molecular Glue-like Degradation via Recruitment of DCAF16. *ACS Chem Biol* **18**, 331-339, doi:10.1021/acscchembio.2c00747 (2023).
- 4 Donovan, K. A. *et al.* Mapping the Degradable Kinome Provides a Resource for Expedited Degradation Development. *Cell* **183**, 1714-1731 e1710, doi:10.1016/j.cell.2020.10.038 (2020).
- 5 Adhikari, B. *et al.* PROTAC-mediated degradation reveals a non-catalytic function of AURORA-A kinase. *Nat Chem Biol* **16**, 1179-1188, doi:10.1038/s41589-020-00652-y (2020).
- 6 Khan, Z. M. *et al.* Structural basis for the action of the drug trametinib at KSR-bound MEK. *Nature* **588**, 509-514, doi:10.1038/s41586-020-2760-4 (2020).
- 7 Gibson, W. J. *et al.* Bifunctional small molecules that induce nuclear localization and targeted transcriptional regulation. *bioRxiv*, doi:10.1101/2023.07.07.548101 (2023).
- 8 Ouyang, W. *et al.* Analysis of the Human Protein Atlas Image Classification competition. *Nat Methods* **16**, 1254-1261, doi:10.1038/s41592-019-0658-6 (2019).
- 9 Filippakopoulos, P. *et al.* Selective inhibition of BET bromodomains. *Nature* **468**, 1067-1073, doi:10.1038/nature09504 (2010).
- 10 Skowronek, P. *et al.* Rapid and In-Depth Coverage of the (Phospho-)Proteome With Deep Libraries and Optimal Window Design for dia-PASEF. *Mol Cell Proteomics* **21**, 100279, doi:10.1016/j.mcpro.2022.100279 (2022).
- 11 Scholl, F. A. *et al.* Mek1/2 MAPK kinases are essential for Mammalian development, homeostasis, and Raf-induced hyperplasia. *Dev Cell* **12**, 615-629, doi:10.1016/j.devcel.2007.03.009 (2007).
- 12 Gerhard, D. S. *et al.* The status, quality, and expansion of the NIH full-length cDNA project: the Mammalian Gene Collection (MGC). *Genome Res* **14**, 2121-2127, doi:10.1101/gr.2596504 (2004).

REVIEWERS' COMMENTS

Reviewer #2 (Remarks to the Author):

The authors have done significant work to address the concerns. I am satisfied with their response and support publication.

Reviewer #3 (Remarks to the Author):

The authors have addressed my concerns to my satisfaction, and I now endorse publication in Nature Communications.